# API Pack: A Massive Multi-Programming Language Dataset for API Call Generation

**Zhen Guo** [*†]
MIT EECS
zguo0525@mit.edu

**Adriana Meza Soria** [*]
MIT-IBM Watson AI Lab
adriana.meza.soria@ibm.com

**Wei Sun**
IBM Research
sunw@us.ibm.com

**Yikang Shen**
MIT-IBM Watson AI Lab
yikang.shen@ibm.com

**Rameswar Panda**
MIT-IBM Watson AI Lab
rpanda@ibm.com

## Abstract

We introduce API Pack, a massive multi-programming language dataset containing over one million instruction-API calls for improving the API call generation capabilities of large language models. Our evaluation highlights three key findings: First, fine-tuning on API Pack enables open-source models to outperform GPT-3.5 and GPT-4 in generating code for entirely new API calls. We show this by fine-tuning CodeLlama-13B on 20,000 Python instances from API Pack. Second, fine-tuning on a large dataset in one language, combined with smaller datasets from others, improves API generation accuracy across multiple languages. Third, we confirm the benefits of larger datasets for API generalization, as increasing fine-tuning data to one million instances enhances generalization to new APIs. To support further research, we open-source the API Pack dataset, trained model, and code at https://github.com/zguo0525/API-Pack.

## 1 Introduction

Large language models (LLMs) have shown promise in assisting in software engineering tasks (Li et al., 2023a; Ozkaya, 2023; Fan et al., 2023; Belzner et al., 2023; Ross et al., 2023; Hou et al., 2023; Ebert & Louridas, 2023; Chang et al., 2024; Wang et al., 2024), with a primary focus on code generation (Chen et al., 2021b; Xu et al., 2022; Sarsa et al., 2022; Vaithilingam et al., 2022; Poesia et al., 2022; Wang et al., 2023b; Shrivastava et al., 2023a; Liang et al., 2023; Zan et al., 2023; Liu et al., 2023a; Shrivastava et al., 2023b; Wei et al., 2023; Muennighoff et al., 2023; Li et al., 2023c; Huang et al., 2023; Lozhkov et al., 2024; Thakur et al., 2024; Mishra et al., 2024). While these advances are significant, developers still face a time-consuming challenge: manually finding and adapting API code examples from lengthy documentation (Meng et al., 2018). Although prior research has addressed API intent detection (finding the right API for a task), the more complex challenge of generating specific API call code has received limited attention, particularly given the varying syntax of HTTP protocol elements across programming languages.

To achieve this goal, we create API Pack, a dataset designed to improve the API call generation capabilities of LLMs. With over 1 million instances spanning 10 programming languages, API Pack represents the largest open-source instruction dataset for API call generation and intent detection (see Table 1). Our dataset distinguishes itself from prior work (Zan et al., 2022; Zhang et al., 2023a; Xu et al., 2023b; Patil et al., 2023; Qin et al., 2023) in two key aspects: programming language diversity and dataset size. By including API calls across multiple languages, API Pack enables the first comprehensive study of cross-language skill transfer in API generation, as it examines how improvements in one language generalize to others. Additionally, its extensive collection of real-world API examples allows for the evaluation of generalization capabilities through controlled variations in training data volume.

---

[*]These authors contributed equally to this work.
[†]Work performed while at MIT-IBM Watson AI Lab. Currently at Apple AIML.

Table 1: A comparison of API Pack with other instruction datasets for API intent detection and/or API call code generation. The upper section of the table reports the features that each dataset covers, and the bottom section reports the data statistics available (/ means unavailable).

| Feature | API Pack (this work) | APIBench (Gorilla) | ToolBench | ToolBench (ToolLLM) | API Bank | ToolAlpaca | ToolFormer |
|---|---|---|---|---|---|---|---|
| API call intent detection? | ✓ | ✓ | ✓ | ✓ | ✓ | ✓ | ✓ |
| API call code generation? | ✓ | ✓ | ✓ | ✗ | ✗ | ✗ | ✗ |
| Multi-programming language? | ✓(10) | ✗(Python) | ✓(Curl, Python) | ✗ | ✗ | ✗ | ✗ |
| Multi-API call scenario? | ✗ | ✗ | ✓ | ✓ | ✓ | ✓ | ✓ |
| Data generation method | custom | self-instruct | self-instruct | custom | custom | custom | custom |
| # of Sources | 4 | 3 | 8 | 1 | 53 | / | 5 |
| # of APIs / Tools | 11,213 | 1,645 | 8 | 16,464 | 53 | 400 | 5 |
| # of API calls | 1,128,599 | 16,450 | / | 37,204 | 568 | 3,938 | 9,400 |
| # of Instances | 1,128,599 | 16,450 | 2,746 | 12,657 | 264 | 3,938 | 22,453 |

We summarize three key findings from our experiments:

1. Fine-tuning CodeLlama-13B on 20k Python API Pack instances outperforms GPT-3.5 by 10% and GPT-4 by 5% on unseen API calls.

2. Cross-language API call generation can be enabled by a large amount of data in one programming language plus small amounts of data in others.

3. Increasing fine-tuning data from 0 to 1 million instances improves generalization to new APIs, demonstrating clear benefits of larger datasets for API generalization.

This paper is structured as follows: We begin with a review of related work (Section 2), followed by a detailed description of the API Pack dataset construction (Section 3). Our experimental method for model fine-tuning and evaluation is presented in Section 4, with key findings discussed in Section 5. We conclude in Section 6 with a discussion of limitations and future directions. Additional considerations regarding ethics and broader impact are addressed in Appendices A.1 and A.2, respectively. The API Pack dataset is made available under the Creative Commons Attribution 4.0 International License, with accompanying source code released under the MIT License.

## 2    RELATED WORK

### 2.1    METHODS TO GENERATE INSTRUCTION DATA WITH LLMS

The manual creation of instruction datasets is a time-consuming and resource-intensive process (Xu et al., 2023a). To address this challenge, researchers have developed automated methods leveraging LLMs. Two notable approaches in this domain are Self-Instruct (Wang et al., 2023a) and Evol-Instruct (Xu et al., 2023a), both focused on generating and filtering instruction-response pairs. Self-Instruct utilizes in-context learning to modify seed examples and employs ROUGE-L similarity and heuristics for filtering. In contrast, Evol-Instruct uses targeted prompting for data generation and relies on predefined heuristics for filtering.

For code-specific datasets, hybrid LLM-based pipelines offer an alternative approach. These pipelines typically combine high-quality human code samples with LLM-generated instructions (Patil et al., 2023; Xu et al., 2023b). While human verification remains the gold standard Wang et al. (2023a), it becomes impractical at scale. Recent work has instead adopted powerful LLMs like GPT-4 and Mistral as automated judges, using example-guided heuristics derived from human-reviewed data (Liu et al., 2023c). Similar techniques have been applied to evaluate instruction complexity (Chen et al., 2023; Lu et al., 2023).

### 2.2    LLMS FOR API CALL CODE GENERATION AND INTENT DETECTION

LLMs for code generation have predominantly focused on general coding tasks, as evaluated by benchmarks like MBPP (Austin et al., 2021), HumanEval (Chen et al., 2021a), and their variants (Liu et al., 2023b; Zheng et al., 2023; Peng et al., 2024). Recent work has expanded into two specific API-related domains: API call intent detection and API call code generation.

API call intent detection focuses on matching natural language tasks to appropriate API endpoints. LLMs designed for this purpose (Zan et al., 2022; Zhang et al., 2023a; Qin et al., 2023; Li et al., 2023b; Tang et al., 2023; Yang et al., 2023; Schick et al., 2023) often operate in hybrid architectures, where the LLM identifies the relevant API endpoint(s) while other components handle the code generation. These studies have explored both single-API and multi-API intent detection. Achieving a strong performance in the latter remains a challenge Qin et al. (2023). At present, this is likely the way most function calling LLMs have been trained to operate (Srinivasan et al., 2023; Yuan et al., 2024; Chen et al., 2024).

Our work, however, focuses on training LLMs for API call code generation, which remains under explored compared to API call intent detection. To the best of our knowledge, only two projects have previously worked on this challenge: Gorilla (Patil et al., 2023), which generates API calls to load pre-trained machine learning models from known model hubs, and ToolBench (Xu et al., 2023b), which focuses on improving LLM's tool manipulation capabilities for a few applications.

## 3 API PACK

API Pack[1] is an instruction dataset with over one million instances. Each instance contains an input-output pair plus additional information about the API and respective endpoints. The inputs are instructions for finding an API call to solve a coding task, including a task description in software engineering language and the name of the API. The outputs are API call examples, specifically HTTP request code snippets curated from OpenAPI specification (OAS) files. The data in API Pack is sourced from four hubs that store OAS files: RapidAPI[2], APIGurus[3], Swaggerhub[4], and IBM's public API Hub[5]. To the best of our knowledge, these API Hubs were the only sources with publicly available OAS files at a large scale at the time we collected data. From these sources, we kept all the APIs that passed our filtering criteria (see Section 3.1). Table 2 summarizes the total number of APIs, unique endpoints, and total instances that are part of API Pack.

Table 2: Final count of data curated per source, where each instance contains one API call

| Source | APIs | Unique Endpoints | Total Instances |
|---|---|---|---|
| RapidAPI | 4,115 | 21,525 | 270,095 |
| APIs Gurus | 1,980 | 37,097 | 495,533 |
| SwaggerHub | 5,045 | 26,747 | 345,765 |
| IBM API Hub | 73 | 2,884 | 17,206 |
| **Total** | **11,213** | **88,253** | **1,128,599** |

The construction of API Pack involves four main stages: data pre-processing (Section 3.1), API Database (DB) creation (Section 3.2), instruction generation (Section 3.3), and data validation (Section 3.4). Figure 1 illustrates the overall pipeline.

### 3.1 DATA PRE-PROCESSING

This process involves two main steps. First, we filter out OAS files with non-English metadata (less than 1%) and those with zero endpoints, as they cannot be used to generate API calls. Next, we extract key information from the remaining OAS files. At the API level, we collect the name, description, and provider. At the endpoint level, we extract the name, functionality, description, method, and path. To ensure data quality, we remove instances missing critical information needed for generating API calls (e.g., method, path, endpoint name) or instructions (e.g., functionality, description). This step ensures that all dataset instances include the necessary details for generating accurate API call examples and instructions.

---

[1]https://huggingface.co/datasets/apipack/API-Pack-Dataset

[2]https://rapidapi.com/categories

[3]https://apis.guru/

[4]https://app.swaggerhub.com/search

[5]https://developer.ibm.com/apis/

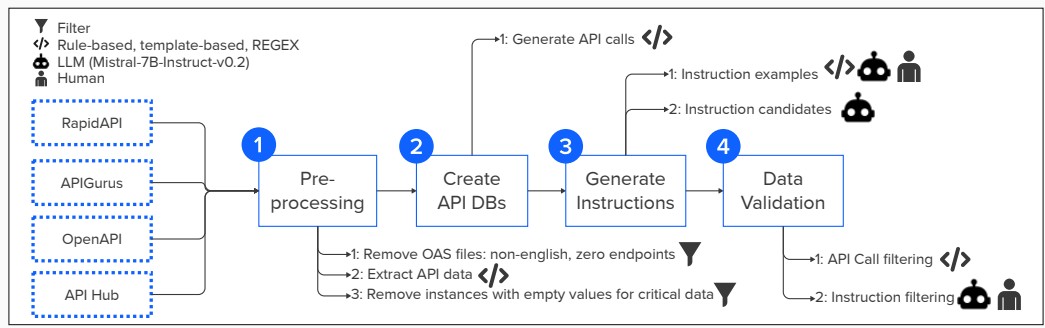

Figure 1: Dataset curation pipeline.

## 3.2 CREATE API DATABASES

We construct an API database (DB) from the pre-processed data, where each instance contains endpoint and API details. Using the OpenAPI Snippet[6] library, we generate API calls in 10 programming languages (cURL, libcurl, Java, Node.js, Python, Go, Ruby, PHP, Swift, and JavaScript) for each instance. This process is applied to data from RapidAPI, APIGurus, and OpenAPI. For IBM API Hub, we directly extract API calls from OAS files, as these already include calls in multiple languages. The selection of programming languages is based on two factors: (1) the 10 languages align with those highlighted in the StarCoder report (Li et al., 2023c), and (2) the OpenAPI Snippet library's support for these languages. Appendix A.7 provides details on the API DB structure and language diversity across sources.

## 3.3 INSTRUCTIONS GENERATION

This process involves creating high-quality instruction examples and generating instruction candidates based on them. First, we randomly select three endpoints from each API DB and use their details (e.g., functionality, description, endpoint name, path) along with the API name to fill predefined instruction templates. This results in three instruction examples per API DB file. For the API Gurus and IBM API Hub sources, three co-authors of this work manually refine the generated instruction examples. The refinement criteria are: 1) correcting grammatical errors, 2) removing unnecessary information, and 3) ensuring the API name is included and correct in the examples. Most annotations were done independently, but annotators discussed non-standard cases to reach consensus. To streamline the refinement process for Swaggerhub and RapidAPI sources, we use an LLM (Mistral-7B-Instruct-v0.2[7]) instead of human effort. We create a prompt (see Prompt 2 in Appendix A.11) based on manual review heuristics and use it to task the LLM with refining instruction examples for these sources. Next, we use the high-quality instruction examples to generate five instruction candidates for each API DB instance. We prompt the LLM with the endpoint's information and provide the high-quality instruction examples as in-context examples (see Prompt 3 in Appendix A.11). Figure 7 in Appendix A.8 shows an instance with one instruction candidate and its corresponding API call. Appendix A.9 presents all five candidates generated for the same instance.

## 3.4 DATA VALIDATION

The data validation process involves three main steps: 1) verifying the validity of API calls as HTTP request examples in a given programming language, 2) assessing the quality of generated instructions, and 3) selecting the highest-quality instruction for fine-tuning.

First, to verify API calls, we ensure that the endpoint_name and HTTP method (e.g., GET, POST) in the API call match the instance data. We use regular expressions to validate the URL format, accounting for placeholders in the domain, path parameters, or query parameters. Additionally,

---

[6]https://www.npmjs.com/package/openapi-snippet
[7]https://huggingface.co/mistralai/Mistral-7B-Instruct-v0.2

we check that the programming language keywords in the API call correspond to the language ID assigned to each instance.

Second, to assess instruction quality, we randomly sampled 121 instances, each with one API call and five instructions, totaling 605 instructions. A co-author with expertise in software development manually classified these instructions as good or bad, with a second annotator providing feedback. Bad instructions typically: 1) contain multiple instructions instead of one, 2) include unnecessary text, or 3) incorrectly use the API name. Based on these traits, we created three prompts with fixed in-context examples to automatically annotate instructions using an LLM. We used the LLM (Mistral-7B-Instruct-v0.2) to annotate the 605 instructions and compared the results with human annotations. We selected the prompt with the best performance (see Prompt 5 in Appendix A.11) and used it to classify all instructions in our dataset. We then removed instances with fewer than two good instructions.

Third, to select the best instruction candidate from the good ones, we calculated the likelihood that an LLM would regenerate the input text used to create the instruction. We prompted Mistral-7B-Instruct-v0.2 with each instruction candidate (see Prompt 4 in Appendix A.11) and obtained the log probability of each token for the regenerated input text. We computed the mean log probability (`input_tokens_mean`) and selected the instruction annotated as good with the highest `input_tokens_mean` for fine-tuning. Our final dataset includes 1,128,599 instances, each with a valid API call and at least two high-quality instructions. Appendix A.3 details the data instances filtered out at each pipeline stage.

## 4 EXPERIMENTS AND EVALUATION FRAMEWORK

To optimize the instruction-following capabilities of the language models, we post-process API Pack into two instruction-tuning templates. The **0-shot-tuning** template targets scenarios where the output is expected to be a straightforward inference from the given input. The **3-shot-tuning** template, with random selections within the selected API, emphasizes the model's ability to learn and generate output with in-context learning. The **3-shot-tuning** template is available in Appendix 6. Model generation parameters follow the BigCodeBench (Zhuo et al., 2024).

### 4.1 EXPERIMENTAL SETTINGS

**A. Selecting a base model:** Our first experimental study serves the purpose of selecting a base model for the rest of our experiments. We fine-tune Mistral 7b (Jiang et al., 2023), CodeLlama 7b and 13b, as well as Llama 2 13b (Touvron et al., 2023) on a subset of API Pack (20,000 instances in Python programming language). We evaluate the performance of resulting models and select the best-performing base model.

**B. Inference with retrieval:** Our second experiment aims to understand the influence of retrieval augmentation on model generalization. We evaluate the models under five distinct prompt settings during test time:

- **0-shot**: No API examples provided for the model.
- **3-shot random**: Three randomly selected API examples.
- **3-shot retrieved**: Three retrieved relevant API examples.
- **3-shot retrieved & re-ranked**: Five retrieved API examples, with three selected using a re-ranker model in the final prompt.
- **3-shot oracle**: Three retrieved relevant API examples. Replacing one of the randomly selected retrieved API examples with the oracle example.

The **3-shot random inference** uses the same prompt template as **3-shot-tuning**. We use *bge-large-en-v1.5* (Zhang et al., 2023b) as the embedding model for retrieval and *bge-reranker-large* (Xiao et al., 2023) for re-ranking the API examples. Appendix A.6 includes more details of the inference pipeline used to evaluate the models' performance.

**C. Cross-language generalization:** To test the model's ability to generalize to new programming languages, we supplement a cURL dataset of 100,000 instances with 1,000 instances from each of the

9 additional languages in API Pack: Go, Java, JavaScript, libcurl, Node.js, PHP, Python, Ruby, and Swift. The goal is to determine if a model can generalize to new languages without requiring a large amount of multi-programming language data. Notably, these instances across different languages invoke the same API functionality, allowing us to evaluate how well models can translate the same API invocation across different programming languages.

**D. Scaling experiment:** We conduct a scaling experiment to investigate whether more API data improves a model's generalization ability for unseen APIs. We fine-tune models on progressively larger API datasets with unique API calls on 0k, 10k, 20k, 40k, 80k, 100k, and 1 million instances. Our hypothesis is that exposure to a greater scale and diversity of APIs during fine-tuning will improve the model's ability to generalize to new, unseen APIs.

## 4.2 Evaluations

To measure the generalization capabilities enabled by API Pack, we establish a comprehensive evaluation framework spanning three levels of complexity for API call generation:

- **Level 1**: Seen APIs and endpoints. This level assesses generalization to new instructions.
- **Level 2**: Seen APIs and new endpoints. This level tests generalization to new endpoints of known APIs.
- **Level 3**: Unseen APIs and endpoints. This level validates performance on entirely new APIs.

The process of data splitting is detailed in Appendix A.4, and training hyperparameters are provided in Appendix A.5. To evaluate the accuracy of the generated Endpoints and API calls, we use the `SequenceMatcher` class from Python's `difflib` library. This algorithm computes a similarity ratio between two sequences by finding the longest contiguous matching subsequences while considering certain elements as insignificant. The similarity ratio $r$ is calculated as:

$$r = \frac{2M}{T},$$

where $M$ is the total number of matching characters (excluding insignificant elements), and $T$ is the total number of characters in both sequences (Black, 2021). Insignificant elements refer to parts of the API requests that do not affect functionality, such as whitespace, formatting differences (e.g., spaces, tabs, newlines), and variable naming conventions. A generated output is considered correct if the similarity ratio meets or exceeds a heuristic threshold of 0.9.

This evaluation focuses on structural and semantic correctness rather than exact syntactic matches. While execution-based metrics like pass rate (Chen et al., 2021b) or hallucination rate (Jain et al., 2024) could offer more robust evaluation, our dataset prioritizes privacy by removing personally identifiable information (PII) from the API database, making execution-based evaluation challenging. We discuss these evaluation limitations and potential future directions in Section 6.

## 5 Experimental Results

### 5.1 Fine-tuned CodeLlama Excels in API Call Generation

We conducted a comprehensive evaluation comparing models fine-tuned on 20,000 Python instances from API Pack against GPT-3.5 and GPT-4. The evaluation assessed fine-tuned open-source models across all three difficulty levels, while GPT-3.5 and GPT-4 were evaluated only on Level 3 due to fine-tuning cost constraints. Table 3 presents these results, demonstrating that open-source LLMs fine-tuned on API Pack can match or exceed the performance of leading proprietary models in generating unseen API calls.

Our analysis reveals three key findings:

1. **Advantage of CodeLlama-13b:** The fine-tuned CodeLlama-13b model consistently outperforms other models, achieving the highest API call accuracy across all evaluation levels, particularly in the 3-shot retrieved setting. For Level 3, it achieves 49.5% accuracy, compared to 42.5% for Mistral-7b and 44.2% for Llama-2-13b.

2. **3-shot vs. 0-shot Training:** Models fine-tuned with the 3-shot-tuning template consistently outperform those trained with the 0-shot-tuning template. For example, CodeLlama-13b's Level 3 accuracy improves from 44.1% (0-shot training) to 49.5% (3-shot training) when using 3-shot retrieved prompts.

3. **Outperforming GPT Models:** Our fine-tuned CodeLlama-13b model surpasses both GPT-3.5 and GPT-4 on Level 3 evaluations. In the 3-shot retrieved setting, CodeLlama-13b achieves 49.5% API call accuracy, compared to 39.5% for GPT-3.5 and 44.3% for GPT-4. This 5.2 percentage point improvement over GPT-4 demonstrates the effectiveness of our fine-tuning approach with API Pack.

Table 3: Evaluation for models fine-tuned with 20k Python API dataset, including a comparison with (not-fine-tuned) CodeLlama-13b, GPT-3.5, and GPT-4. Levels 1 and 2 are not reported for GPT-3.5 and GPT-4 due to proprietary fine-tuning processes. Note that 3-shot (retre) is short for 3-shot retrieved.

| Model | Fine-tuning template | Testing | Evaluation Accuracy (%) | | | | | |
|---|---|---|---|---|---|---|---|---|
| | | | Level 1 | | Level 2 | | Level 3 | |
| | | | Intent | API Call | Intent | API Call | Intent | API Call |
| Mistral-7b | 0-shot | 0-shot | 17.2 | 10.9 | 14.1 | 11.4 | 14.3 | 11.2 |
| | | 3-shot (retre) | 42.0 | 29.7 | 35.4 | 28.7 | 39.1 | 29.1 |
| | 3-shot | 0-shot | 40.5 | 28.5 | 24.0 | 18.3 | 15.2 | 12.1 |
| | | 3-shot (retre) | **64.1** | 55.4 | 49.1 | 42.8 | 50.8 | 42.5 |
| CodeLlama-7b | 0-shot | 0-shot | 8.1 | 6.1 | 10.0 | 7.0 | 11.0 | 7.8 |
| | | 3-shot (retre) | 52.6 | 42.6 | 43.6 | 35.9 | 50.2 | 40.1 |
| | 3-shot | 0-shot | 12.1 | 9.3 | 13.7 | 10.2 | 16.8 | 13.0 |
| | | 3-shot (retre) | 60.6 | 52.7 | 54.1 | 47.3 | 55.9 | 49.1 |
| Llama-2-13b | 0-shot | 0-shot | 9.4 | 6.2 | 11.6 | 9.0 | 10.9 | 8.4 |
| | | 3-shot (retre) | 44.5 | 33.9 | 45.4 | 35.6 | 46.7 | 39.1 |
| | 3-shot | 0-shot | 15.7 | 10.2 | 14.0 | 11.2 | 11.7 | 9.6 |
| | | 3-shot (retre) | 59.5 | 51.5 | 50.8 | 44.3 | 52.7 | 44.2 |
| CodeLlama-13b | 0-shot | 0-shot | 9.8 | 6.8 | 10.8 | 8.1 | 12.1 | 8.5 |
| | | 3-shot (retre) | 55.6 | 44.4 | 50.6 | 43.3 | 52.3 | 44.1 |
| | 3-shot | 0-shot | 14.4 | 10.3 | 15.9 | 13.3 | 14.2 | 8.9 |
| | | 3-shot (retre) | 63.5 | **55.5** | **56.8** | **51.4** | **56.1** | **49.5** |
| | none | 0-shot | 0.1 | 0.0 | 0.2 | 0.0 | 0.1 | 0.0 |
| | | 3-shot (retre) | 49.2 | 46.2 | 36.3 | 34.4 | 40.7 | 38.5 |
| gpt-3.5-1106 | none | 0-shot | - | - | - | - | 1.0 | 0.7 |
| | none | 3-shot (retre) | - | - | - | - | 47.2 | 39.5 |
| gpt-4-1106 | none | 0-shot | - | - | - | - | 0.2 | 0.1 |
| | none | 3-shot (retre) | - | - | - | - | 53.5 | 44.3 |

## 5.2 RETRIEVAL AUGMENTATION IMPROVES API CALL GENERATION

Table 4: Performance comparison for different retrieval methods in 3-shot prompting. Note that 3-shot (retre & rerank) is short for 3-shot retrieved & re-ranked.

| Model | Testing | Evaluation Accuracy (%) | | | | | |
|---|---|---|---|---|---|---|---|
| | | Level 1 | | Level 2 | | Level 3 | |
| | | Endpoint | API Call | Endpoint | API Call | Endpoint | API Call |
| Mistral-7b | 3-shot (rand) | 54.5 | 41.8 | 48.2 | 41.2 | 45.2 | 37.0 |
| | 3-shot (retre) | 64.1 | 55.4 | 49.1 | 42.8 | 50.8 | 42.5 |
| | 3-shot (retre & rerank) | 63.0 | 53.6 | 49.0 | 42.2 | 51.5 | 43.9 |
| | 3-shot (oracle) | 75.8 | 68.4 | 62.5 | 56.8 | 60.2 | 53.1 |
| CodeLlama-13b | 3-shot (rand) | 49.2 | 38.6 | 49.8 | 43.6 | 50.0 | 41.4 |
| | 3-shot (retre) | 63.5 | 55.5 | 56.8 | 51.4 | 56.1 | 49.5 |
| | 3-shot (retre & rerank) | 61.0 | 52.9 | 55.1 | 49.2 | 55.9 | 49.3 |
| | 3-shot (oracle) | 78.5 | 71.2 | 69.8 | 64.1 | 65.9 | 59.3 |

Table 4 shows how different retrieval methods affect 3-shot API call generation. Our analysis highlights four key findings:

1. **Advantages of Retrieved Examples:** 3-shot (retre) consistently outperforms 3-shot (rand) for both Mistral-7b and CodeLlama-13b models across all evaluation levels.

2. **Limitations of Generic Re-ranking:** Interestingly, 3-shot (retre & rerank) does not consistently improve upon 3-shot (retre) and sometimes performs slightly worse (e.g., Level 1 drops from 63.5% to 61.0% for CodeLlama-13b). Our 3-shot (oracle) results show that better example selection can improve performance. However, the current re-ranker fails to capture these gains, suggesting that a fine-tuned re-ranker adapted to API characteristics might be more effective.

3. **Performance Gap Across Levels:** We observe a notable performance gap between retrieval methods for Level 1, which narrows for Levels 2 and 3. This trend suggests that retrieval is particularly effective for seen APIs and endpoints (Level 1).

4. **Endpoint vs. API Call Accuracy:** Across all conditions, Endpoint accuracy is consistently higher than API Call accuracy. This indicates that identifying the correct endpoint is easier than generating the full API call, which requires accurately specifying more parameters, their types, and any required formatting.

These results demonstrate that at inference time, 3-shot (retre) is the most effective approach for fine-tuned Mistral-7b and CodeLlama-13b models compared to 3-shot (rand) and 3-shot (retre & rerank).

## 5.3 CROSS-LANGUAGE GENERALIZATIONS IN API CALL PERFORMANCE

Figure 2 compares three fine-tuning approaches for cross-language API call performance: a model fine-tuned exclusively on 100,000 instances of 'cURL' data, or CURL MODEL, three EXPERT MODELS, each fine-tuned on 100,000 samples of 'cURL', 'Python', and 'Java' data separately, and a MIXTURE MODEL fine-tuned on 100,000 instances of 'cURL' data with additional samples of 1,000 instances each for nine different languages. Since all these instances invoke the same API functionality across different languages, this comparison directly evaluates the models' ability to translate equivalent API calls across programming languages.

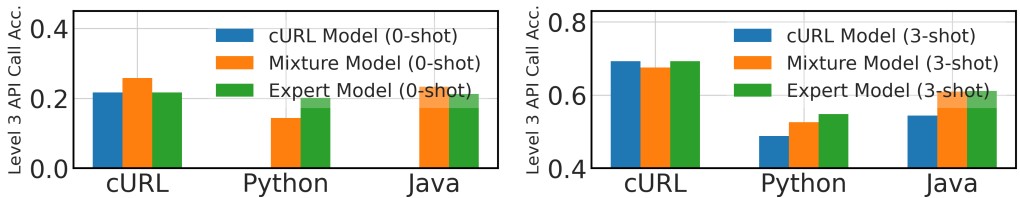

Figure 2: Comparison of 0-shot and 3-shot API call performance for different models in cURL, Python, and Java. Note that the EXPERT MODELS are specific to each programming language.

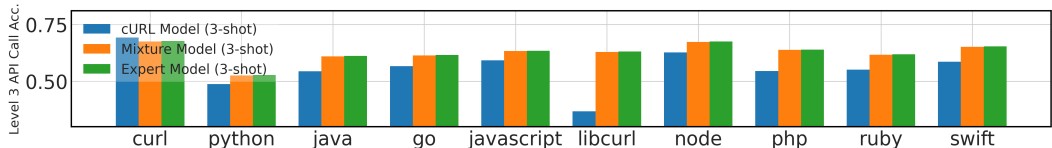

Figure 3: Three-shot performance in ten languages across different models.

The comparsion shows that without fine-tuning, models struggle to generalize to new programming languages in 0-shot scenarios. However, in 3-shot settings, performance improves significantly, demonstrating the benefits of in-context learning.

Figure 3 illustrates that the MIXTURE MODEL performs comparably to EXPERT MODELS across ten programming languages in 3-shot testing. This demonstrates the MIXTURE MODEL's ability to generalize effectively without significant performance loss, highlighting its robustness and versatility for multi-language programming applications.

Overall, these findings indicate that cross-language API call generation is achievable by fine-tuning models with a large dataset in one language and smaller datasets in others, eliminating the need for extensive data in each target programming language.

## 5.4 SCALING INSTRUCTION DATASET HELPS GENERALIZATION

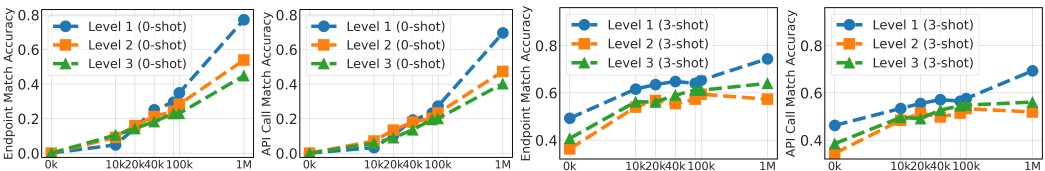

Figure 4: Scaling instruction dataset, with 3-shot fine-tuning template, on CodeLlama-13b with 0-shot and 3-shot retrieval evaluations. The x-axis represents the log scale size of fine-tuning data from API Pack. The y-axis is Endpoint or API call accuracy.

Figure 4 shows that 0-shot performance increases with the fine-tuning dataset size (ranging from 0k for the non-finetuned baseline to 1 million instances from API Pack using a 3-shot fine-tuning template). We observe a roughly linear rise in both endpoint match accuracy and API call match accuracy as the dataset grows. This trend is consistent across all evaluation levels (1, 2, and 3), indicating that larger datasets enhance the model's ability to generate API calls without examples, likely due to exposure to more diverse API patterns and structures. For 3-shot prompting, the graph also shows performance gains as the dataset size increases, though these gains are smaller than in the 0-shot case, especially for level 3 cases.

For Level 1 (seen APIs and endpoints), there is a similar linear improvement as in the 0-shot case, but with a gentler slope. This suggests that even with in-context examples, larger fine-tuning datasets further improve performance on familiar APIs. However, for Levels 2 (new endpoints of known APIs) and 3 (unseen APIs), performance initially improves but then plateaus as fine-tuning data approaches 1 million instances. The plateau is most evident for Level 3, our most challenging evaluation scenario. These findings indicate that there may be diminishing returns from scaling instruction data, particularly for the most difficult generalization tasks (Level 3).

## 5.5 ADDITIONAL EXPERIMENTS

We conducted two experiments to assess the benefits of API Pack: First, to evaluate the impact of using multiple API sources, we fine-tuned CodeLlama-13b on 20k instances from ToolBench, which only includes APIs from RapidAPI. Second, to test our data filtering pipeline, we created a Non-filtered API Pack by replacing 45% of the filtered instructions by the LLM with non-filtered ones, and fine-tuned CodeLlama-13b on 20k instances.

Table 5: Performance comparison for ToolBench, Non-filtered API Pack, and API Pack in different shot settings.

| Training Dataset | Testing | Evaluation Accuracy (%) | | | | | |
| --- | --- | --- | --- | --- | --- | --- | --- |
| | | Level 1 | | Level 2 | | Level 3 | |
| | | Endpoint | API Call | Endpoint | API Call | Endpoint | API Call |
| ToolBench | 0-shot | 5.7 | 5.7 | 8.1 | 8.0 | 7.3 | 7.0 |
| | 3-shot | 44.5 | 37.8 | 40.7 | 36.4 | 43.7 | 38.1 |
| Non-filtered API Pack | 0-shot | 10.3 | 8.3 | 12.8 | 10.3 | 12.2 | 8.4 |
| | 3-shot | 62.4 | 54.9 | 54.9 | 48.7 | 55.7 | 49.5 |
| API Pack | 0-shot | 14.4 | 10.3 | 15.9 | 13.3 | 14.2 | 8.9 |
| | 3-shot | **63.5** | **55.5** | **56.8** | **51.4** | **56.1** | 49.1 |

Table 5 presents the results of these experiments alongside our API Pack performance. Key findings from these experiments include:

1. **Multi-source Advantage**: Training with APIs from a single source (ToolBench) reduces performance in both 0-shot and 3-shot settings compared to API Pack, which aggregates data from four sources. This demonstrates the benefit of our multi-source approach.

2. **Data-filtering Importance**: Introducing non-filtered instructions (Non-filtered API Pack) leads to a slight performance drop compared to the fully filtered API Pack. This indicates that while our filtering process contributes to dataset quality, its overall impact on performance appears limited, suggesting room for simplification or refinement in future iterations.

We also evaluated the performance of mixing a subset of 50,000-entries of API Pack with Magicoder dataset and fine-tuned CodeLlama-13b model. The resulting model shows an increase of over 35.3% in API call code generation accuracy for Level 3, specifically with the 3-shot setting. This improvement does not come at the expense of general coding efficiency, as the resulting model still performs well on benchmarks such as HumanEval+ and MBPP. See Table 6 for further details.

Table 6: Evaluation for Code Generation with CodeLlama-13b

| Data Mixture | Bench. (pass@10) | | Level 3 (3-shot) |
| --- | --- | --- | --- |
| | HumanEval+ | MBPP | Endpoint |
| - | 47.8 | 58.3 | - |
| Magicoder | 60.8 | 66.4 | 17.0 |
| Magicoder + API Pack | 61.3 | 64.3 | 52.3 |

## 6 CONCLUSION AND FUTURE WORK

We introduced API Pack, a multi-programming language dataset containing over 1 million instruction-API call pairs to transform software development workflows by enabling LLMs to accurately generate API calls from natural language instructions. Our experimental results demonstrated three key achievements: First, fine-tuning CodeLlama-13B on 20,000 Python instances from API Pack enables it to outperform GPT-3.5 and GPT-4 in generating unseen API calls. Second, we established that model performance continues to improve as training data scales to 1 million instances, assisting generalization to unseen APIs. Third, we showed that cross-language API call generation can be achieved by combining extensive training data in one programming language with smaller amounts from other languages.

We acknowledge that the current API Pack exhibits several limitations. Our evaluation method, while systematic, could benefit from execution-based metrics like pass rates or hallucination rates, but our focus on privacy protection and PII removal currently makes such testing difficult. Additionally, API Pack was not designed for multi-API call scenarios, and its requirement for explicit API name specification may limit application in complex software development scenarios. Programming language coverage is constrained by code generation library limitations, potentially excluding languages of interest to end-users. Furthermore, while API-gurus operates under a CC0-1.0 license, the limited availability of API-level license information may restrict use in proprietary development contexts.

Looking ahead, we plan to address these challenges through several initiatives. We will explore privacy-preserving execution-based evaluation methods using synthetic endpoints or sanitized environments. Our roadmap includes expanding the dataset to support multi-API scenarios and task-based instructions, while also increasing programming language diversity. Finally, we aim to improve API-level license documentation to better serve the needs of all potential users.

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

## A  APPENDIX

### A.1  ETHICAL AND CONSIDERATIONS

Developers and other kind of end-users that plan to use API Pack for LLM fine-tuning should consider the following factors:

- **API Evolution and Validity:** APIs included in API Pack may evolve over time, potentially rendering some data outdated. Users should be aware that fine-tuning models on static API datasets might introduce deprecated or incorrect API calls. We recommend periodically updating the dataset or employing retrieval-augmented generation techniques to ensure the model has access to the most recent API specifications.

- **Hallucination:** API Pack's code (API calls) was not generated via an LLM. Thus, there is a low risk of code errors in the dataset due to LLMs hallucination. That said, hallucination is a general risk associated with the use of LLMs. Therefore, code generated with LLMs that were fine-tuned with API Pack or any other code dataset must be always verified by a developer.

- **Data Ownership and License Information:** The OAS files used to build API Pack come from public and official API data hubs. API data owners published these files on those websites themselves. If a legitimate API code owner requires their data to be removed from API Pack we will do so. Regarding API's license, only few OAS files contained this information. We are committed to tracking down complete and accurate license information for all APIs included in API Pack in a future version of the dataset.

- **Sensitive Information:** There is no risk of sensitive information being released as API Pack code data was sourced from API documentation that does not include real argument values. Instead, placeholders indicate the need for an argument value (i.e., RE-PLACE_BASIC_AUTH). In a real application, the users can provide these values through a conversational AI interface (i.e., code assistant). Some API calls do include public and restricted API URLs, however these where directly released by the API owners.

## A.2 Broader Impact

API Pack enables the integration of code LLMs specialized in API call generation into software development processes, which could have significant broader impacts:

- **Enhanced Software Productivity:** The presented advancements promise significant software development acceleration by automating routine coding tasks. This could enable faster iterations, reduced cost, and potentially increase the quality of products. However, reliance on auto-generated code risks reduced developer accountability and control. The oversight of human eye to ensure accuracy is still required.

- **Sociotechnical Considerations and Responsible Innovation:** Although intelligent coding tools enhance productivity, sociotechnical concerns exist regarding job displacement, ownership, and digital inequality. We design API Pack to leverage the creation of tools that increase software productivity by keeping humans in the loop, as developers still decide whether or not using the API calls generated with models fine tuned on API Pack. Regarding code ownership, we consider API Pack not to have a negative effect, as data is sourced from documentation that is already publicly available. Towards reducing digital inequality, we have decided to open-sourced our dataset, code, and models.

## A.3 Data Filtering at each Stage of the Pipeline

Table 7 shows the number of instances from each API source at different stages of our data filtering pipeline. Detailed description for each stage can be found in Section 3.

Table 7: Data Filtering Progress

| Source/Instances | Before Data Validation | After Removing Invalid API calls | After Removing Instances without Good Instructions |
|---|---|---|---|
| IBM API Hub | 27,635 | 17,712 | 17,206 |
| APIs Gurus | 500,160 | 499,250 | 495,533 |
| Swaggerhub | 351,756 | 351,756 | 345,765 |
| RapidAPI | 274,014 | 273,388 | 270,095 |
| **Total** | **1,153,565** | **1,142,106** | **1,128,599** |

## A.4 Data Spliting for Level 1, Level 2, and Level 3 Testing

To split the data, we first copy the total dataset into total_training and calculate the occurrence of each API. For the first level of testing data, select up to 1,000 data points from the first 20,000 entries where instruction_test is not "None" and the API appears more than four times. For the second level, iterate through the first 20,000 entries, selecting data points with APIs not yet selected, until 1,000 unique APIs are collected, then remove these from the training dataset. For the third level, from the 80,000th entry onwards, select data points with APIs not used in Level 2, collecting up to 1,000 data points, and remove data points with these APIs from the training dataset.

### A.5 HYPERPARAMETERS FOR TRAINING

We fine-tune the models using the HuggingFace Transformers library on a cluster consisting of 1 node with 8 NVIDIA H100 80GB GPUs, with Fully Shared Data Parallelism (FSDP). Techniques such as mixed precision, gradient checkpointing, and AdaFactor optimizer are used to improve training efficiency. The key hyperparameters are summarized in Table 8.

Table 8: Hyperparameters for Training

| Hyperparameter Name | Value |
| --- | --- |
| Learning rate | $2 \times 10^{-5}$ |
| Batch size | 128 |
| Max seq length | 4096 |
| Number of epochs | 2 |
| Warmup ratio | 0.03 |

### A.6 TESTING PIPELINE

Figure 5 outlines our testing pipeline for 0-shot and few-shot learning. The retriever uses an embedding model with cosine similarity to find Top-k semantically similar examples from the example collection. A reranker then refines and re-orders these examples. The LLM generates the appropriate API call based on the user input and retrieved examples. For 0-shot inference, no examples are included in the prompt. For few-shot learning, we use a 3-shot approach, incorporating the three most relevant examples. The prompt template is in Listing 6.

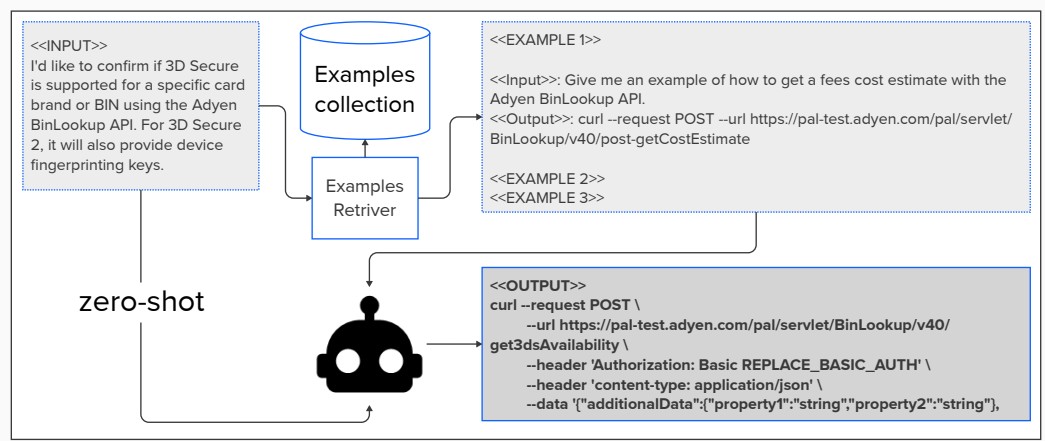

Figure 5: Testing pipeline.

## A.7 API DB INSTANCE

Figure 6 shows the structure of an API DB instance. We used openapi-snippet [8], an open-source package that takes as input an OpenAPI v2.0 or v3.0.x specification file (OAS file) and translates it into an HTTP Archive 1.2 request object, to generate API calls. We generated API calls (api_call) in 10 different programming languages (cURL, libcurl, java, node, python, go, ruby, php, swift, JavaScript) for RapidAPI, API Gurus, and the Swaggerhub. For the IBM Developer Hub API calls were extracted directly from the OAS files. We extracted API calls in eight different programming languages from this source (cURL, java, node, python, go, ruby, php, swift).

```
{
    "api_name" : "The name of the API the endpoint belongs to",
    "api_description": "The description of the API the enpoint
    ↪  belongs to",
    "api_provider": "The API's provider name",
    "endpoint_name": "The name of the function to call",
    "functionality": "A brief description of endpoint's
    ↪  functionality",
    "description": "A long description of endpoint's functionality",
    "path": "The enpoint's name plus specific versioning as it
    ↪  appears in the API call's URL",
    "method": "HTTP method used in the API call (e.g., get, post, put,
    ↪  delete)",
    "api_call": "HTTP request code example to invoke an endpoint",
    "lang": "The programming language in which the API call is
    ↪  written",
}
```

Figure 6: Structure of an API DB instance

## A.8 API PACK DATASET INSTANCE EXAMPLE

Figure 7 shows a fragment of an API Pack data instance. As data is sourced from public OAS files, personally identifiable and sensitive information in API calls is represented by anonymous strings (e.g., REPLACE_BASIC_AUTH), which indicates the need to pass this value as an argument in real execution. In this example, we only show one instruction candidate for clarity.

```
{
    "instruction_candidates": [
        {
            "idx": 1,
            "candidate": "I'd like to confirm if 3D Secure is supported for a specific
            ↪  card brand or BIN using the Adyen BinLookup API. For 3D Secure 2, it will
            ↪  also provide device fingerprinting keys.",
            "input_tokens_mean": -0.5497539341557909,
        },
        ...
    ],
    "api_call": "curl --request POST \
    --url https://paltest.adyen.com/pal/servlet/BinLookup/v40/get3dsAvailability \
    --header 'Authorization: Basic REPLACE_BASIC_AUTH' \
    --header 'content-type: application/json' \
    --data '{"additionalData":{"property1":"string","property2":"string"},
            "brands":["string"],
            "cardNumber":"string",
            "merchantAccount":"string",
            "recurringDetailReference":"string",
            "shopperReference":"string"}'"
    ...
}
```

Figure 7: Fragment of an API Pack instance in cURL.

[8]https://www.npmjs.com/package/openapi-snippet

## A.9 INSTRUCTION CANDIDATES

Figure 8 shows all the candidates generated for an instance with their respective id and input_tokens_mean.

```
"instruction_candidates": [
    {
        "idx": 1,
        "candidate": "I'd like to confirm if 3D Secure is supported
        ↪  for a specific card brand or BIN using the Adyen
        ↪  BinLookup API. For 3D Secure 2, it will also provide
        ↪  device fingerprinting keys.",
        "input_tokens_mean": -0.5497539341557909
    },
    {
        "idx": 2,
        "candidate": "To confirm if 3D Secure is enabled for a
        ↪  specific card brand or BIN number using the Adyen
        ↪  BinLookup API, you can make use of the
        ↪  `post-get3dsAvailability` endpoint. This API call will
        ↪  provide you with information about the availability of 3D
        ↪  Secure, along with device fingerprinting keys for 3D
        ↪  Secure 2 transactions. Let me know if you need assistance
        ↪  in forming the request or handling the response.",
        "input_tokens_mean": -0.5088001283229344
    },
    {
        "idx": 3,
        "candidate": "To confirm if a specific card brand or BIN
        ↪  supports 3D Secure and retrieves device fingerprinting
        ↪  keys for 3D Secure 2, please utilize the Adyen BinLookup
        ↪  API's post-get3dsAvailability endpoint.",
        "input_tokens_mean": -0.555471563123543
    },
    {
        "idx": 4,
        "candidate": "To confirm if 3D Secure is supported for a
        ↪  specific card brand or BIN number using the Adyen
        ↪  BinLookup API, please make a POST request to the endpoint
        ↪  'post-get3dsAvailability'. This API will return whether
        ↪  3D Secure is enabled and for 3D Secure 2, it will also
        ↪  provide device fingerprinting keys.",
        "input_tokens_mean": -0.5466722401065375
    },
    {
        "idx": 5,
        "candidate": "To confirm if 3D Secure is supported by Adyen
        ↪  for a given card brand or BIN number, you can utilize the
        ↪  Adyen BinLookup API. Simply send a POST request to the
        ↪  post-get3dsAvailability endpoint with the required card
        ↪  details. If 3D Secure is available, the response will
        ↪  include device fingerprinting keys for 3D Secure 2.",
        "input_tokens_mean": -0.5726057469087047
    }
]
```

Figure 8: Examples of five instruction candidates generated for the same instance.

## A.10 QUALITY ASSESSMENT OF COLLECTED INSTRUCTIONS

We recognize that utilizing a smaller, open-sourced model like Mistral-7B for data generation might raise concerns about the quality of the instructions produced. To address these concerns, we conducted an evaluation to demonstrate the generality and reliability of our collected datasets. Specifically, we used GPT-4o-mini to assess the quality of the instructions in our API pack test dataset.

In this evaluation, we used a detailed prompt (Listing 1) to guide gpt-4o-mini in categorizing the instructions into High, Medium, or Low quality based on their naturalness and clarity. The assessment revealed that the majority of instructions were rated as either High or Medium quality, affirming the dataset's suitability for training purposes.

Figure 9 illustrates the distribution of instruction quality across the three different levels. As shown, each level maintains a substantial proportion of high-quality instructions, with Level 1 exhibiting the highest percentage of High-quality ratings. The distributions of Medium and Low-quality instructions provide insights into areas where further refinements can be made.

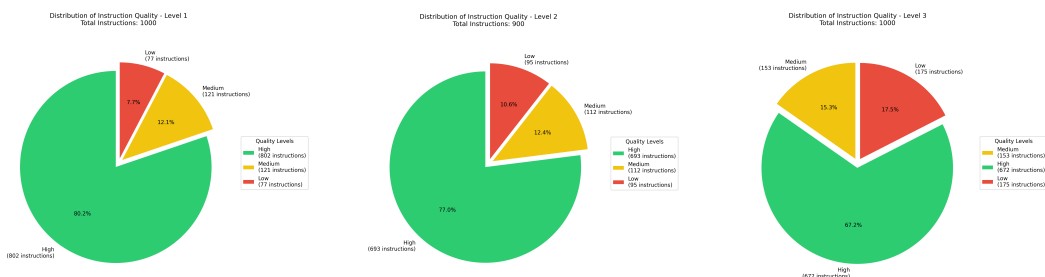

Figure 9: Instruction data quality distribution for Level 1, 2, and 3 test set.

Listing 1: Instruction Quality Evluation Prompt for gpt-4o-mini

```
**Prompt:**

Evaluate the naturalness of the `instruction_test` in the given data.
    Assign a quality label of **High**, **Medium**, or **Low** based on
    how naturally a user would ask the question when seeking to use the
    specified API endpoint. Include detailed reasoning for your
    conclusion. Use clean, simple, and concise language.

---

**Examples:**

**Example 1 (High Quality):**

**Data:**
```json
{
  "api_name": "Weather Service",
  "api_call_data": {
    "functionality": "Get current weather by city",
    "description": "Retrieve the current weather information for a
    specified city.",
    "path": "/weather/current/{city}"
  },
  "instruction_test": "How can I get the current weather for a specific
    city using the Weather Service?"
}
```

**Evaluation:**
- **Label:** High
```

- **Reasoning:** The instruction is clear and naturally phrased. It
  directly asks how to get current weather information using the API,
  making it easy to understand.

---

**Example 2 (High Quality):**

**Data:**
```json
{
  "api_name": "Library API",
  "api_call_data": {
    "functionality": "Search books by author",
    "description": "Find books written by a specific author.",
    "path": "/books/author/{authorName}"
  },
  "instruction_test": "How do I search for books by a particular author
   using the Library API?"
}
```

**Evaluation:**
- **Label:** High
- **Reasoning:** The instruction is naturally worded and clearly conveys
  the user's intent to search for books by author using the API.

---

**Example 3 (Medium Quality):**

**Data:**
```json
{
  "api_name": "Movie Database",
  "api_call_data": {
    "functionality": "Get movie details",
    "description": "Retrieve details about a movie by its ID.",
    "path": "/movies/{movieId}"
  },
  "instruction_test": "Explain how to get details of movie using Movie
   Database."
}
```

**Evaluation:**
- **Label:** Medium
- **Reasoning:** The instruction is understandable but slightly awkward.
  It lacks the article "a" before "movie" and doesn't mention using the
   API explicitly.

---

**Example 4 (Medium Quality):**

**Data:**
```json
{
  "api_name": "Flight Info Service",
  "api_call_data": {
    "functionality": "Check flight status",
    "description": "Get the status of a flight using its flight number.",
    "path": "/flights/status/{flightNumber}"
  },
```

```
    "instruction_test": "How to use Flight Info Service to check status of
        flight?"
}
```

**Evaluation:**
- **Label:** Medium
- **Reasoning:** The instruction is somewhat clear but could be more
    natural. It misses the article "the" before "status" and "a" before "
    flight," making it slightly less fluent.

---

**Example 5 (Low Quality):**

**Data:**
```json
{
  "api_name": "Music Streaming API",
  "api_call_data": {
    "functionality": "Play a song",
    "description": "Stream a song by its ID.",
    "path": "/songs/play/{songId}"
  },
  "instruction_test": "Need play song Music Streaming API."
}
```

**Evaluation:**
- **Label:** Low
- **Reasoning:** The instruction is ungrammatical and lacks proper
    sentence structure, making it unnatural and hard to understand.

---

**Example 6 (Low Quality):**

**Data:**
```json
{
  "api_name": "Stock Market API",
  "api_call_data": {
    "functionality": "Get stock price",
    "description": "Retrieve the current price of a stock by its ticker
    symbol.",
    "path": "/stocks/{tickerSymbol}/price"
  },
  "instruction_test": "Price stock get API."
}
```

**Evaluation:**
- **Label:** Low
- **Reasoning:** The instruction is fragmented and lacks coherence. It
    doesn't form a complete, understandable question.

---

**Now, evaluate the following data:**

**Data:**
```json
{
  "api_name": "{api_name}",
  "api_call_data": {
```

```
    "functionality": "{functionality}",
    "description": "{description}",
    "path": "{path}"
  },
  "instruction_test": "{instruction_test}"
}
```

**Evaluation:**
- **Label:**
- **Reasoning:**

## A.11 PROMPTS

In this section, we share all the prompts used to create instructions. We also explain the process to craft each prompt. When in use, the values within the curly brackets would be replaced by the actual data.

### A.11.1 EXAMPLES REFINEMENT

We use Prompt 2 to refine the examples for instruction generation. We created three examples per API. Examples were manually refined by a human for the APIs from IBM API Hub and APIs Gurus. Then, we tasked an LLM with refining the examples for the other two sources.

Listing 2: Prompt for instruction example refinement

```
Your task is to refine and enhance a user query that involves a specific
    API. The original query you'll work with includes key details about
    the API's functionality, description, endpoint, and name. Focus on
    these essential aspects when revising the query:

1. **Integration of API Details:** Make sure the revised query includes
    relevant details about the API's functionality, description, and name
    , without directly mentioning the endpoint.

2. **Grammar and Syntax Correction:** Analyze the original query for
    grammatical mistakes such as improper verb forms (e.g., 'can'
    followed by 's' or 'es') or misplaced punctuation (like commas or
    colons). Correct these to improve clarity and professionalism.

3. **Relevance and Conciseness:** Eliminate any extraneous information
    from the original query. Strive for brevity while ensuring all
    critical details are included.

4. **User-Centric Rewrite:** Rework the query to reflect a user's
    perspective, focusing on their specific needs and how the API can
    address those needs.

For each query, you will receive:

- **Input:** An original user query with API details.
- **Your Task:** Revise the query based on the guidelines above.

Example for Practice:

### Input:
Functionality: Search Ecards
Description: Allows searching the inventory system for ecards.
Endpoint: searchECards
API: eCards Search API
User query to refine: "Please tell me how to searches ecards with the
    eCards Search API."

### Output (refined user query):
```

```
"Can you guide me on how to search for ecards using the eCards Search API
    ?"

Another Example:

### Input:
Functionality: KPI of realized sales
Description: Provides KPIs for documents issued, requires ACLs from /
    instances, and uses the 'typeKpi' parameter.
Endpoint: getKPIs
API: Blackbird Analytics
User query to refine: "I need to certificates shows kpis related to all
    documents issued through blackbird. to use it you must have pass a
    list of acls retrieved by /instances and specify which kpi using '
    typekpi' parameter."

### Output (refined user query):
"How can I access KPIs for documents issued by Blackbird Analytics, and
    what are the required 'typeKpi' parameters?"

Remember, the goal is to modify the user query to be clear, effective,
    and grammatically correct, fully showcasing how the user can leverage
     the specific API.

Now, here is your actual task:
### Input:
Functionality: {functionality}
Description: {description}
Endpoint: {endpoint}
API: {api_name}
User query to refine: {template generated instruction}

### Output (refined user query):
```

### A.11.2 INSTRUCTION GENERATION

We use Prompt 3 to generate five instruction candidates for each API Pack instance. We crafted a custom prompt that considers the information a developer may provide to an LLM, while looking for an API call to complete a task.

Listing 3: Prompt for instruction generation

```
Your task is to create a user query that effectively utilizes a specific
    API. The API's functionality, description, and name will be provided
    to you. Your query should be designed in a way that makes the best
    use of this API's unique capabilities. When crafting your query,
    focus on:

1. **API Name Integration:** Clearly include the API's name in your query
    to ensure relevance.
2. **Specificity:** Replace broad or vague terms with precise, concrete
    details relevant to the API's purpose.
3. **Conciseness:** Keep your query as brief as possible while still
    fully conveying the needed information. Avoid unnecessary verbosity.
4. **Excluding API Endpoint:** Do not include the API's endpoint in your
    query; focus only on the user's need and how the API fulfills it.

Create a query that a user might realistically use when interacting with
    the given API. Think about typical scenarios or problems that the API
     is designed to solve and formulate your query accordingly.

Examples for practice:

###Input:
```

```
Functionality: {functionality}
Description: {description}
Endpoint: {endpoint}
API: {api_name}
###Output:
{output}

###Input:
Functionality: {functionality}
Description: {description}
Endpoint: {endpoint}
API: {api_name}
###Output:
{output}

###Input:
Functionality: {functionality}
Description: {description}
Endpoint: {endpoint}
API: {api_name}
###Output:
{output}

Remember, the goal is to demonstrate how a user would benefit from this
    specific API in a realistic scenario, using precise and clear
    language. Here is the actual task for you:

###Input:
Functionality: {functionality}
Description: {description}
Endpoint: {endpoint}
API: {api_name}
###Output:
```

### A.11.3 BACK TRANSLATION

Prompt 4 seeks to re-generate the API information associated to an instruction by passing an instruction candidate as input. To create this prompt, we took Prompt 3 as reference and simply reversed the task. As a result of using this prompt for inference, we save the input-token-mean of the output return (the API information that was re-generated).

Listing 4: Prompt for instruction backtranslation

```
Your task involves a reverse-engineering process where you will analyze a
    user query to infer specific details about an API endpoint. Based on
    the given user query, you are expected to:

1. **Identify the Endpoint's Identifier:** Derive the endpoint identifier
    that aligns with the functionality implied by the user query.
2. **Determine Endpoint Functionality:** Interpret the user query to
    understand and describe the functionality of the endpoint.
3. **Describe the Endpoint:** Provide a detailed description of the
    endpoint based on the needs and context presented in the user query.
4. **Specify the API Name:** Identify and state the name of the API to
    which this endpoint belongs, as suggested by the user query.

Your response should clearly articulate these four elements (identifier,
    functionality, description, API name) in a manner that reflects an
    accurate understanding of the user query. Consider the query as a
    real-world scenario or problem that the endpoint is designed to
    address.

Examples for practice:
```

```
###Input:
{generated instruction}
###Output:
Functionality: {functionality}
Description: {description}
Endpoint: {endpoint}
API: {api_name}

###Input:
{generated instruction}
###Output:
Functionality: {functionality}
Description: {description}
Endpoint: {endpoint}
API: {api_name}

###Input:
{generated instruction}
###Output:
Functionality: {functionality}
Description: {description}
Endpoint: {endpoint}
API: {api_name}

The goal is to showcase your ability to connect a user's needs with the
    appropriate API endpoint, demonstrating an understanding of how the
    endpoint's features align with user requirements. Your response
    should be precise, insightful, and reflective of the query's
    implications. Here is the actual task for you:

###Input:
{generated instruction}
###Output:
Functionality: {functionality}
Description: {description}
Endpoint: {endpoint}
API: {api_name}
```

### A.11.4 INSTRUCTION ANNOTATION

We use Prompt 5 to task an LLM with annotating the instruction candidates as good or bad. First, a human annotator manually reviewed 605 instructions to distill common error patterns among them. Then, we craft a prompt to detect these error patterns via an LLM. Note the prompt includes examples for 'good' and 'bad' instructions. These examples were extracted from the manual revision performed by the human annotator.

Listing 5: Prompt for instruction annotation

```
**Your Task**: Evaluate the provided instruction from an AI assistant and
    classify its quality as **Good** or **Bad** based on specific
    criteria.

**Criteria for 'Bad' Instruction**:
1. Contains multiple instructions instead of a single, clear directive.
2. Includes unnecessary additional text before or after the main
    instruction.
3. Fails to accurately use the specified API name and endpoint.

**Input Structure**:
You will receive an **INPUT** consisting of three elements:
1. An **instruction** generated by the AI assistant.
2. The **API name** related to the instruction.
3. The **API endpoint** relevant to the instruction.
```

```
**Output**:
Classify the instruction as **Good** or **Bad** with a concise
    justification.

**Examples**:

1. **Input**:
   - **Instruction**: "Create a new message in IBM Event Streams using
    the REST Producer API." "How do I format and send a message body via
    the IBM Event Streams REST Producer API?" "Can you help me construct
    a message using the IBM Event Streams REST Producer?" "Use the IBM
    Event Streams REST Producer API to send a message with specific data
    ." "What's the proper syntax for creating and sending a message
    through the IBM Event Streams REST Producer?"
   - **API Name**: IBM Event Streams REST Producer
   - **API Endpoint**: produceMessage
   - **Output**: Bad. This instruction is classified as **Bad** because
    it combines four separate instructions into one, each asking for
    different guidance related to the IBM Event Streams REST Producer API
    .

2. **Input**:
   - **Instruction**: "I'd like to send a new message to IBM Event
    Streams. How do I format the request body to ensure it is correctly
    processed using the IBM Event Streams REST Producer API?"
   - **API Name**: IBM Event Streams REST Producer
   - **API Endpoint**: produceMessage
   - **Output**: Good. This instruction is classified as **Good** because
     it provides a single, clear directive without additional text and
    correctly uses the API name.

3. **Input**:
   - **Instruction**: "Here's a possible user query utilizing the given
    API: 'Help me list all the bare metal servers in my Virtual Private
    Cloud account using the Virtual Private Cloud API.'"
   - **API Name**: Virtual Private Cloud API
   - **API Endpoint**: list_bare_metal_servers
   - **Output**: Bad. This instruction is classified as **Bad** because
    it contains unnecessary introductory text ("Here's a possible user
    query utilizing the given API:") before the actual instruction.

4. **Input**:
   - **Instruction**: "How do I retrieve a list of all bare metal servers
     in my region using the Virtual Private Cloud API?"
   - **API Name**: Virtual Private Cloud API
   - **API Endpoint**: list_bare_metal_servers
   - **Output**: Good. This instruction is classified as **Good** because
     it is a singular, straightforward instruction without extra text and
    appropriately uses the API name.

**Your Current Task**:

**Input**:
- **Instruction**: {candidate}
- **API Name**: {api_name}
- **API Endpoint**: {endpoint}

**Output**:
```

A.11.5   FEW-SHOT FINE-TUNING AND EVALUATION

This is the prompt template used for 3-shot-tuning to post-process API Pack data, as well as for 3-shot random, 3-shot retrieved, and 3-shot retrieved & re-ranked inference.

Listing 6: Prompt for few-shot fine-tuning and evaluation

```
**api_description**:{api_description}
**lang**:{programming language}

Given the following examples:

**instruction**
{instruction}
**output**
{api_call}

**instruction**
{instruction}
**output**
{api_call}

**instruction**
{instruction}
**output**
{api_call}

Your actual task:
**instruction**
{instruction_test}
**output**

### ASSISTANT:
```

