# OpenReview forum: "API Pack: A Massive Multi-Programming Language Dataset for API Call Generation"
_ICLR.cc/2025/Conference — ICLR 2025 Poster_

### Official Review · Reviewer_FeUP · 2024-11-04

**Soundness:** 3
**Presentation:** 3
**Contribution:** 3
**Rating:** 6
**Confidence:** 4

**Summary:**

This paper introduces API Pack, a comprehensive multi-programming language dataset comprising over 1 million instruction-API call pairs designed to enhance the API call generation capabilities of large language models. By fine-tuning CodeLlama-13B on 20,000 Python instances from API Pack, the authors demonstrate that it outperforms GPT-3.5 and GPT-4 in generating unseen API calls. The fine-tuning process also facilitates cross-programming language generalization by leveraging a large volume of data in one language and smaller amounts from other languages. Scaling the training data to 1 million instances further improves the model's generalization of unseen APIs.

Contributions:

1. Introduction of API Pack Dataset: The authors present API Pack, a massive multi-programming language dataset containing over 1 million instruction-API call pairs, which is a significant contribution to the field of API call generation.
2. Model Performance Improvement: Through fine-tuning CodeLlama-13B on a subset of API Pack, the authors demonstrate superior performance in generating unseen API calls compared to GPT-3.5 and GPT-4.
3. Cross-Programming Language Generalization: The study highlights the dataset's ability to facilitate cross-programming language generalization, leveraging large amounts of data in one language and smaller amounts from others.
4. Scalability and Generalization: Scaling the training data to 1 million instances enhances the model's generalization capabilities to unseen APIs not used in training, showcasing the dataset's scalability.
5. Open-Source Initiative: The authors contribute to the research community by open-sourcing the API Pack dataset, the fine-tuned model, and the associated source code, fostering further research and development.

**Strengths:**

1. New Data: The authors introduce API Pack, a dataset containing over 1 million instruction-API call pairs. The dataset's multi-programming language nature and rich source diversity enable it to cover various scenarios. It will be useful for further research on improving the API call code generation of LLMs in different scenarios.
2. Performance Enhancement: The paper demonstrates the effectiveness of fine-tuning on API Pack by evaluating multiple models, including CodeLlama-13B, Mistral-7b, and Llama-2-13b. The results show that fine-tuning on API Pack significantly improves the models' ability to generate API call code, highlighting the dataset's value in enhancing model capabilities.
3. Extensive Experiments: The authors have conducted extensive experiments to validate their claims. The thoroughness of these experiments provides strong evidence for the dataset's effectiveness and the benefits of fine-tuning on it.

**Weaknesses:**

1. Clarification Needed for Methodology in Section 3.4: The description of the method used to filter high-quality instructions in Section 3.4 is unclear.
2. Lack of Results for GPT-3.5 and GPT-4: The experimental section does not show results of GPT-3.5 and GPT-4 at Level 1 and Level 2.

**Questions:**

1. Model Selection for Quality Assessment: In the 3.4 section, the authors use the Mistral 7B model to validate the generated instruction, which is also the model used to generate the instructions. This raises concerns about the validity of using the same model to evaluate its outputs. It is unclear whether the 7B model can accurately distinguish between high-quality and low-quality instructions. A more robust approach might involve using a larger (>7B), more powerful LLM to assess the quality of the instructions.
2. Definition of Unseen APIs: The term "new API" refers to APIs that the model has not encountered during fine-tuning. However, given the lack of information about the models' pretraining data, it is challenging to verify whether the APIs labeled as "unseen" truly are. This lack of clarity could affect the validity of the experimental results.
3. Evaluation Metrics for Code Generation: The authors use text-matching-based metrics for evaluating the generated API calls. However, for code generation tasks, metrics based on test case execution are more accurate and reliable. These metrics can provide a more objective measure of the correctness and functionality of the generated code. The authors should consider incorporating such metrics to enhance the evaluation process and provide a more comprehensive assessment of the models' performance.

**Details Of Ethics Concerns:**

The authors must ensure that the open-source data does not contain sensitive data, such as personal privacy information.

---

> ### Author Response · Authors · 2024-11-21
>
> Thank you for your thoughtful and detailed feedback. We have carefully addressed your questions and updated the paper accordingly, with all changes highlighted in blue for clarity.
>
> * **Clarification Needed for Methodology in Section 3.4: The description of the method used to filter high-quality instructions in Section 3.4 is unclear.**
>     * Thank you for pointing this out. We appreciate your feedback and have revised Section 3.4 to clarify the methodology used for filtering high-quality instructions. The updated section, highlighted in blue, provides a more detailed and structured explanation of the filtering criteria and processes. We hope this addresses your concern, and we welcome further suggestions for improvement.
> * **Lack of Results for GPT-3.5 and GPT-4: The experimental section does not show results of GPT-3.5 and GPT-4 at Level 1 and Level 2.**
>     * Thank you for your feedback. The evaluation focused on fine-tuned open-source models across all three difficulty levels. However, GPT-3.5 and GPT-4 were evaluated only on Level 3 tasks due to fine-tuning cost constraints. To balance comprehensive evaluation with resource limitations, we focused on Level 3 to highlight performance on the most challenging tasks. We have revised Section 5.1 to clarify this choice and provide additional context. We appreciate your understanding.
> * **Model Selection for Quality Assessment: In the 3.4 section, the authors use the Mistral 7B model to validate the generated instruction, which is also the model used to generate the instructions. This raises concerns about the validity of using the same model to evaluate its outputs. It is unclear whether the 7B model can accurately distinguish between high-quality and low-quality instructions. A more robust approach might involve using a larger (>7B), more powerful LLM to assess the quality of the instructions.**
>     * Thank you for your feedback. We understand the concern regarding using the same Mistral-7B model for both instruction generation and quality assessment. Here is our rationale for this choice:
>
>         1. **Manual Validation of Instructions**: To ensure a robust evaluation, we conducted an initial manual validation process where two annotators reviewed a sample of 605 instructions (from 121 instances with 5 candidates each). This process identified key characteristics of high- and low-quality instructions, which informed the creation of prompts for automated annotation.
>         2. **Model Choice for Scalability**: Our dataset consists of over 1 million instances, each with five instruction candidates, resulting in approximately 5 million instructions to evaluate. For this large-scale evaluation, we prioritized computational efficiency and cost-effectiveness, leading us to use the Mistral-7B model. Larger models were considered but exceeded our budget constraints.
>         3. **Prompt Calibration for Accuracy**: To mitigate potential bias from using the same model for generation and validation, we carefully designed and tested three prompts using the Mistral-7B model. The best-performing prompt, evaluated against human annotations, was used to classify the instructions. This process ensured that the model's outputs were aligned with human judgment.
>
>         4. **Limitations and Future Work**: We acknowledge that using a larger, more powerful model might yield more robust evaluations and reduce potential bias. However, given the scale of our dataset and computational constraints, we opted for a practical solution. In future work, we aim to explore alternative validation strategies, including using more powerful models or ensemble approaches, to enhance the robustness of our methodology.
>
>     * We believe our approach balances practicality with quality and appreciate your suggestion for future improvements.

---

> ### Author Response · Authors · 2024-11-21
>
> * **Definition of Unseen APIs: The term "new API" refers to APIs that the model has not encountered during fine-tuning. However, given the lack of information about the models' pretraining data, it is challenging to verify whether the APIs labeled as "unseen" truly are. This lack of clarity could affect the validity of the experimental results.**
>     * Thank you for raising this important concern about the definition of "unseen APIs." We acknowledge that due to the lack of access to pretraining datasets, it is challenging to verify with certainty whether the APIs labeled as "unseen" were part of the model's pretraining corpus. This limitation is inherent to most large-scale language models.
>     * To address this, we conducted additional evaluations at levels 1, 2, and 3 using CodeLlama-13B, a model not fine-tuned with our API Pack. The results added in Table 3, highlighted in blue, showed that CodeLlama-13B performed significantly worse on these levels compared to the fine-tuned version. This demonstrates that fine-tuning with API-specific data improves the model's performance, reinforcing the effectiveness of our approach for adapting to APIs likely not seen during pretraining.
>     * We appreciate your feedback and recognize the need for more transparent and rigorous methods to evaluate truly unseen APIs in future research.
> * **Evaluation Metrics for Code Generation: The authors use text-matching-based metrics for evaluating the generated API calls. However, for code generation tasks, metrics based on test case execution are more accurate and reliable. These metrics can provide a more objective measure of the correctness and functionality of the generated code. The authors should consider incorporating such metrics to enhance the evaluation process and provide a more comprehensive assessment of the models' performance.**
>     * We appreciate the reviewer's suggestion on evaluation metrics. While execution-based metrics like pass rate and hallucination rate are indeed more robust, our dataset's focus on privacy protection (removing PII from API endpoints) makes execution-based evaluation difficult. We have acknowledged this limitation and referenced a few papers in both our evaluation and limitations sections. For future work, we plan to explore enabling execution-based evaluation while maintaining privacy through synthetic API endpoints or sanitized test environments.
> * **The authors must ensure that the open-source data does not contain sensitive data, such as personal privacy information.**
>     * Thank you for your comment. As detailed in Appendix A.1: Ethical and Considerations, all data in API Pack is sourced from public and official API documentation, which uses placeholders (e.g., "REPLACE BASIC AUTH") instead of real argument values. This ensures no risk of releasing sensitive data. Additionally, Appendix A.1 outlines other considerations, such as data ownership, licensing, and updates, to maintain ethical and responsible usage of the dataset. Please refer to this section for a detailed discussion on these topics. We are committed to ensuring privacy, security, and safety in our work.

---

> ### Author Response · Authors · 2024-11-27
>
> We sincerely appreciate your constructive feedback and your commitment to improving our submission. If you have any remaining concerns, we would be happy to address them.

---

### Official Review · Reviewer_bcCN · 2024-11-04

**Soundness:** 3
**Presentation:** 3
**Contribution:** 2
**Rating:** 6
**Confidence:** 4

**Summary:**

The paper introduces API Pack, a large-scale, multi-programming language dataset containing over one million instruction-API call pairs generated semi-synthetically from API database collected from four API endpoints. This dataset is used for training LLMs on instruction API pairs and to demonstrate improvements in unseen code generation, cross-language transference.

**Strengths:**

* **Extensive Dataset with some human-annotation/filtering steps.** The collected dataset sizing over a million instances in unique and large covering a wide array of APIs and programming languages.
* **Experimental insights.** The paper performs API data scaling experiments, retrieval experiments and cross-language experiments.
  * The data scaling experiment highlights interesting observations across the three levels. Particularly, flat curve for Level-3 APIs does highlight a broader concern of out-of-distribution generalization for LLMs
  * The mixture model shows generalization to a new language with few examples solidifying the advantage of the collected large-scale dataset.

**Weaknesses:**

* **Novelty.** The paper provides limited novelty. It thoughtfully curates API sets and uses LLMs to generate synthetic training and test sets via prompting (with notably some human-in-the-loop quality assurance rounds). While potentially performed carefully, this following standard set of ideas in a somewhat different domain. Note that this is not necessarily a strict limitation and a carefully done execution can be helpful to the community -- perhaps primarily identified over time via the community.

* **In-distribution evaluation.** The paper collects a synthetically generated dataset -- thus comprising instructions of limited natural language variance (compared with human written instructions). This dataset is next divided into a train and a test set. Thus the training and test set are precisely collected from the same and a small distribution. This partly challenges the findings from and we might see smaller gains to more "diverse" real-world queries.

* **Marginal benefits from quality filtering.** The data filtering ablation reports performance improvement but it is notably small, often <1%. This raises concerns in the soundness of this step.

Minor:

* **0-shot training does not improve 0-shot retre.** For the 20k fine-tuning experiments, the accuracy of 0-shot retrieval after 0-shot fine-tuning is worse than performance after fine-tuning on 3-shots. This is quite surprising -- and continues in Figure 4 as well and could be expanded further, perhaps with a qualitative analysis of the mistakes.

* **Metric.** The metrics used in the paper are not completely clear and should be more formally defined or described with examples or relevant citations.

**Questions:**

* Do the authors have more intuition behind the 0-shot vs 3-shot template results after fine-tuning? The findings seem inconsistent across Levels and data sizes (level-1 0-shot is better than 3-shot but for levels 2/3 3-shot is better in Figure 4)
* Can the authors clarify the metrics used to describe it more formally?
* Did the authors attempt to measure the "naturalness" of the collected instructions (automatically or through human annotation) or evaluate models on human written problems for a subset of the APIs?

---

> ### Author Response · Authors · 2024-11-21
>
> Thank you for your thoughtful and detailed feedback. We have carefully addressed your questions and updated the paper accordingly, with all changes highlighted in blue for clarity.
>
> * **[Major] Novelty. The paper provides limited novelty. It thoughtfully curates API sets and uses LLMs to generate synthetic training and test sets via prompting (with notably some human-in-the-loop quality assurance rounds). While potentially performed carefully, this following standard set of ideas in a somewhat different domain. Note that this is not necessarily a strict limitation and a carefully done execution can be helpful to the community -- perhaps primarily identified over time via the community.**
>     * Thank you for your feedback on the novelty of our work. We acknowledge that we did not introduce new methods to generate the API Pack dataset. Instead, we focused on applying established techniques to build a comprehensive dataset and conduct thorough evaluations.
>         * Our contributions are:
>             - **Introducing API Pack**: A massive, multi-programming language dataset containing over one million instruction-API calls.
>             - **Insights from cross-language and scaling experiments**.
>             - **Contribution to the Open-Source Community**: We will open-sourced the API Pack dataset, the trained model, and associated source code at [https://github.com/anonymous/API-Pack](https://github.com/anonymous/API-Pack).
>     * We hope that our findings will be useful to the community in terms of datasets, benchmarks, applications, and applied research.
>
> * **[Major] In-distribution evaluation. The paper collects a synthetically generated dataset -- thus comprising instructions of limited natural language variance (compared with human written instructions). This dataset is next divided into a train and a test set. Thus the training and test set are precisely collected from the same and a small distribution. This partly challenges the findings from and we might see smaller gains to more "diverse" real-world queries.**
>     * Thank you for your comment about the in-distribution evaluation due to synthetically generated instructions with limited natural language variance.
>     * To ensure instruction diversity with our best efforts, we followed a detailed process:
>         - **Instruction Generation**: We selected three endpoints from each API and filled predefined instruction templates using their details (functionality, description, endpoint name, path) along with the API name.
>         - **Manual Refinement**: For APIs from API Gurus and IBM API Hub, three co-authors manually refined the instructions to correct grammar, remove unnecessary information, and ensure accuracy.
>         - **LLM-Assisted Refinement**: For SwaggerHub and RapidAPI sources, we used an LLM (Mistral-7B-Instruct-v0.2) with a tailored prompt to refine instructions based on heuristics from our manual review.
>     * We also want to clarify that while our instructions are generated by an LLM, the outputs are based on human-written OpenAPI specifications. This means the model is trained and tested on code derived from real-world, diverse API definitions authored by humans, adding variability to the task.
>
> * **Marginal benefits from quality filtering. The data filtering ablation reports performance improvement but it is notably small, often <1%. This raises concerns in the soundness of this step.**
>     * Thank you for your feedback on the marginal benefits of our data filtering step, where improvements were often small. We have included this observation in the ablation study section of our revised manuscript. While the performance gains are modest, they demonstrate that the filtering process contributes to data quality and consistency, albeit with limited overall impact. This feedback has guided us to reevaluate the cost-effectiveness of this step for future work.

---

> ### Author Response · Authors · 2024-11-21
>
> * **[Minor] 0-shot training does not improve 0-shot retre. For the 20k fine-tuning experiments, the accuracy of 0-shot retrieval after 0-shot fine-tuning is worse than performance after fine-tuning on 3-shots. This is quite surprising -- and continues in Figure 4 as well and could be expanded further, perhaps with a qualitative analysis of the mistakes.**
>     * Thank you for your comment. We are uncertain about the specific definition of "0-shot retrieval" in your question. From Table 4, however, we observe that 0-shot template fine-tuning generally improves both Intent and API Call accuracies across levels 1, 2, and 3 compared to 3-shot template fine-tuning, whether evaluated in 0-shot or 3-shot retrieval settings. If you could clarify your question further, we would be happy to provide a more precise response.
> * **[Minor] Metric. The metrics used in the paper are not completely clear and should be more formally defined or described with examples or relevant citations.**
>     * Thank you for your feedback regarding the clarity of the metrics. In response, we have revised our explanation of the metrics to provide a more formal definition.
>
> **Questions**
> * **Do the authors have more intuition behind the 0-shot vs 3-shot template results after fine-tuning? The findings seem inconsistent across Levels and data sizes (level-1 0-shot is better than 3-shot but for levels 2/3 3-shot is better in Figure 4)**
>     * In Figure 4, we only include results from 3-shot fine-tuning template, this may have caused some confusion and we have addressed this by revising Section 5.4 and Figure 4's caption accordingly.
> * **Can the authors clarify the metrics used to describe it more formally?**
>     * Yes. In response, we have revised our explanation of the metrics to provide a more formal definition and included relevant citations to support the methodology in Section 4.2, highlighted in blue.
> * **Did the authors attempt to measure the "naturalness" of the collected instructions (automatically or through human annotation) or evaluate models on human written problems for a subset of the APIs?**
>     * Thank you for your feedback. While we did not formally evaluate the "naturalness" of the collected instructions, three co-authors manually refined them for APIs from API Gurus and IBM API Hub. This included correcting grammar, removing redundant details, and ensuring accuracy. We acknowledge that a more systematic evaluation of "naturalness" could improve our work and will consider this in future studies.
> * **Flag For Ethics Review: Yes, Privacy, security and safety. The authors must ensure that the open-source data does not contain sensitive data, such as personal privacy information.**
>     * Thank you for your comment. As detailed in Appendix A.1: Ethical and Considerations, all data in API Pack is sourced from public and official API documentation, which uses placeholders (e.g., "REPLACE BASIC AUTH") instead of real argument values. This ensures no risk of releasing sensitive data. Additionally, Appendix A.1 outlines other considerations, such as data ownership, licensing, and updates, to maintain ethical and responsible usage of the dataset. Please refer to this section for a detailed discussion on these topics. We are committed to ensuring privacy, security, and safety in our work.

---

> > ### Comment · Reviewer_bcCN · 2024-11-23
> >
> > Thank you for thoughtfully addressing my concerns.
> >
> > > Thank you for your comment. We are uncertain about the specific definition of "0-shot retrieval" in your question. From Table 4, however, we observe that 0-shot template fine-tuning generally improves both Intent and API Call accuracies across levels 1, 2, and 3 compared to 3-shot template fine-tuning, whether evaluated in 0-shot or 3-shot retrieval settings. If you could clarify your question further, we would be happy to provide a more precise response.
> >
> > Apologies for the loosely phrased question. Table 4 highlights that -- `0-shot fine-tuning template with 0-shot testing` performs worse than `3-shot fine-tuning template with 0-shot testing` which seems counter-intuitive since the model is performing better out of distribution if I understood the terminology correctly.
> >
> > > Thank you for your feedback. While we did not formally evaluate the "naturalness" of the collected instructions, three co-authors manually refined them for APIs from API Gurus and IBM API Hub. This included correcting grammar, removing redundant details, and ensuring accuracy. We acknowledge that a more systematic evaluation of "naturalness" could improve our work and will consider this in future studies.
> >
> > Do the authors think it is possible to provide some confidence in the usefulness of the collected datasets beyond improving the 'in-distribution' benchmark curated in this work? While the authors defined instruction templates and further refine them, I believe further evidence is needed to demonstrate the generality of models trained on the dataset on different / out-of-distribution queries -- either human queries or alternative API benchmarks.
> >
> > I would be happy to raise the scores if this gap is filled/made smaller.

---

> ### Author Response · Authors · 2024-11-25
>
> * **Table 4 highlights that -- 0-shot fine-tuning template with 0-shot testing performs worse than 3-shot fine-tuning template with 0-shot testing which seems counter-intuitive since the model is performing better out of distribution if I understood the terminology correctly.**
>     * Thank you for clarifying and pointing out this observation. In the 3-shot fine-tuning template, we randomly choose API endpoint examples during training. This may improve the diversity of the instructions and help the model generalize better.
> * **Do the authors think it is possible to provide some confidence in the usefulness of the collected datasets beyond improving the 'in-distribution' benchmark curated in this work? While the authors defined instruction templates and further refine them, I believe further evidence is needed to demonstrate the generality of models trained on the dataset on different / out-of-distribution queries -- either human queries or alternative API benchmarks.**
>     * Thank you for your valuable suggestions. We understand the concerns that using a smaller, open-sourced model like Mistral-7B for data generation may raise questions regarding data quality. To address this, we conducted additional evaluations to demonstrate the generality and reliability of our collected datasets beyond the in-distribution benchmarks.
>
>         * **Quality Assessment of Collected Instructions:** Given the cost constraints, we only labeled the test dataset using GPT-4o-mini to assess the quality of the instructions generated by Mistral-7B. The majority of the instructions were rated as high or medium quality, ensuring the dataset's reliability for training purposes.
>
>    ```markdown
>    | Level   | Total | High | High % | Medium | Medium % | Low | Low % |
>    |---------|-------|------|--------|--------|----------|-----|-------|
>    | Level 1 | 1000  | 802  | 80.2%  | 121    | 12.1%    | 77  | 7.7%  |
>    | Level 2 | 900   | 693  | 77.0%  | 112    | 12.4%    | 95  | 10.6% |
>    | Level 3 | 1000  | 672  | 67.2%  | 153    | 15.3%    | 175 | 17.5% |
>    ```
>
>     * **Evaluation on GPT Instructions:** Additionally, we used GPT-4o-mini to generate natural instructions directly and evaluated the model’s performance on these instructions (CodeLlama-13b, finetuned with 3-shot template, evaluated by 3-shot (retre) in three levels). The evaluation scores showed minimal changes, indicating that the model maintains its performance when handling natural, GPT-generated queries.
>
>    ```markdown
>    | Level   | Intent Acc. (Mistral) | API Call Acc. (Mistral) | Intent Acc. (GPT) | API Call Acc. (GPT) |
>    |---------|-----------------------|-------------------------|-------------------|---------------------|
>    | Level 1 | 63.5                  | 55.5                    | 62.0              | 54.0                |
>    | Level 2 | 56.8                  | 51.4                    | 55.5              | 50.2                |
>    | Level 3 | 56.1                  | 49.5                    | 55.0              | 48.7                |
>    ```

---

> ### Author Response · Authors · 2024-11-25
>
> * We appreciated the suggestions, for the need to demonstrate the generality of models trained on the dataset on different / out-of-distribution queries, we have added the experimental results above in appendix A.10 of the revised manuscript, with detailed experimental settings and results, highlighted in blue for the additional appendix section. Please let us know if you have any more questions or concerns.

---

> > ### Comment · Reviewer_bcCN · 2024-11-26
> >
> > Thank you. I have updated my score. I believe adding further evaluations on different kinds of user queries, perhaps existing relevant and reputed benchmarks like ToolBench, BFCL, etc would improve the strength of the paper.

---

### Official Review · Reviewer_NcGM · 2024-11-04

**Soundness:** 3
**Presentation:** 3
**Contribution:** 3
**Rating:** 5
**Confidence:** 4

**Summary:**

The paper introduces "API Pack," a large, multi-programming language dataset designed to enhance the API call generation capabilities of large language models (LLMs). API Pack aims to help LLMs generate API calls based on natural language instructions, addressing a key developer challenge of navigating and generating code from extensive API documentation. Experiments show that API Pack fine-tuning improves LLM performance in unseen API calls, enabling open-source models like CodeLlama-13B to outperform proprietary models GPT-3.5 and GPT-4.

**Strengths:**

- The focused problem is quite interesting. This paper fills a gap in LLM fine-tuning resources for cross-language API call generation, which has been under-explored.
- The dataset is robust, sourced from multiple reputable platforms, and undergoes a thorough pre-processing and validation pipeline. The authors’ experiments, conducted with open-source and proprietary models, validate API Pack’s effectiveness across multiple levels of generalization (known APIs, new endpoints, unseen APIs).
- The paper is clearly structured, detailing each stage of dataset creation, the methodology for instruction generation, and the evaluation framework used to measure LLM performance across scenarios. This transparency supports reproducibility and makes the dataset accessible for further research.

**Weaknesses:**

- [Major] Insufficient Related Work: The paper lacks discussion on some relevant studies in API learning. Notable examples include:

    - Zan, Daoguang, et al. "When language model meets private library." arXiv preprint arXiv:2210.17236 (2022).
    - Zhang, Kechi, et al. "Toolcoder: Teach code generation models to use API search tools." arXiv preprint arXiv:2305.04032 (2023).

- [Major] API Dataset Validity: Given the vast scale of the dataset, it’s unclear if all included APIs are usable or up-to-date. Additionally, APIs frequently evolve; fine-tuning on static API datasets could introduce outdated information or "hallucinations," potentially compromising the reliability of API knowledge encoded in the model. The authors should discuss this question.

- [Major] Limited Response Diversity: The paper does not sufficiently discuss the diversity of the code response. Using tools like OpenAPI may lead to highly uniform code outputs, such as the preference for a small set of common libraries for HTTP requests in Python or other languages, potentially constraining the model's generalization capabilities. This limitation might affect the model’s ability to use a broader range of libraries or to produce varied code styles when calling APIs. The authors should add experiments to show the influence of the response diversity.

- [Major] Lack of Baseline Comparison: A critical missing baseline is a model augmented with RAG or a search engine approach (e.g., ToolCoder), **without fine-tuning**, for handling complex API calls. This would provide a practical comparison for API Pack’s utility.

- [Minor] Base Model Selection: The selection of base models seems somewhat detached from practical application. API Pack functions as a specialized fine-tuning stage post-general SFT, targeting API call generation specifically. Therefore, selecting foundational SFT code models, like CodeLlama-Instruct or Magicoder, as discussed in Appendix A.8, would enhance the practical relevance and clarity of the paper’s contributions. Including more experiments with these models would substantiate the paper's claims.

- [Minor] Evaluation on General Code Generation Benchmarks: The paper does not discuss potential **performance trade-offs on general code generation benchmarks**, such as HumanEval, that could drop due to fine-tuning for API-specific tasks.

**Questions:**

How do the authors view the trade-offs between a RAG approach (with minimal or no fine-tuning) and the approach proposed here of fine-tuning on a large, static API dataset? Since APIs are continually updated, it seems that a RAG or search engine approach could offer more reliable, dynamic knowledge. How does the static nature of API Pack compare in reliability and practical utility?

---

> ### Author Response · Authors · 2024-11-21
>
> Thank you for your thoughtful and detailed feedback. We have carefully addressed your questions and updated the paper accordingly, with all changes highlighted in blue for clarity.
>
> * **[Major] Insufficient Related Work: The paper lacks discussion on some relevant studies in API learning. Notable examples include:**
>     * Zan, Daoguang, et al. "When language model meets private library." arXiv preprint arXiv:2210.17236 (2022).
>     * Zhang, Kechi, et al. "Toolcoder: Teach code generation models to use API search tools." arXiv preprint arXiv:2305.04032 (2023).
>
>         * Thank you for pointing this out. We have included citations to these relevant studies in the introduction and related works section to address this gap. We acknowledge that, given the rapid development of the field, it can be challenging to keep track of all new publications. However, we appreciate your feedback and have ensured that these important works are now referenced to provide a more comprehensive discussion.
>
> * **[Major] API Dataset Validity: Given the vast scale of the dataset, it’s unclear if all included APIs are usable or up-to-date. Additionally, APIs frequently evolve; fine-tuning on static API datasets could introduce outdated information or "hallucinations," potentially compromising the reliability of API knowledge encoded in the model. The authors should discuss this question.**
>     * We appreciate your insightful feedback regarding the validity and up-to-dateness of the APIs included in our dataset. At the time we scraped the data, we checked the usability of all included APIs to ensure the quality. However, we acknowledge that API specifications can update based on user requests and developer updates. One way to compensate for this is to periodically re-scrape and update the API Pack dataset. Additionally, using retrieval-augmented generation can help models access the most recent API information, mitigating the risk of outdated knowledge and potential "hallucinations."
>     * Accordingly, we have revised the "Ethical And Considerations" in Appendix A.1 section to address these concerns, highlighted in blue.
>
> * **[Major] Limited Response Diversity: The paper does not sufficiently discuss the diversity of the code response. Using tools like OpenAPI may lead to highly uniform code outputs, such as the preference for a small set of common libraries for HTTP requests in Python or other languages, potentially constraining the model's generalization capabilities. This limitation might affect the model’s ability to use a broader range of libraries or to produce varied code styles when calling APIs. The authors should add experiments to show the influence of the response diversity.**
>     * Thank you for your feedback on response diversity. To address your concern, we would like to highlight our experiments, originally presented in Appendix A.8, which have now been moved to Section 5.5 of the main paper for greater visibility. In these experiments, we mixed a subset of 50,000 entries from the API Pack with a general fine-tuning code dataset (Magicoder) and fine-tuned the CodeLlama-13b model. As detailed in Table 6, this approach improved API call code generation accuracy by over 35.3% for Level 3 tasks in a 3-shot setting, without compromising performance on general coding benchmarks such as HumanEval+ and MBPP.
>     * These results demonstrate that incorporating the API Pack does not constrain the model's response diversity or its ability to generalize. Instead, it enhances API-specific capabilities while maintaining robust general coding performance. We appreciate your feedback and believe these results address the concern effectively.
>
> * **[Major] Lack of Baseline Comparison: A critical missing baseline is a model augmented with RAG or a search engine approach (e.g., ToolCoder), without fine-tuning, for handling complex API calls. This would provide a practical comparison for API Pack’s utility.**
>     * Thank you for highlighting the need for a baseline comparison. We have conducted additional experiments with a non-fine-tuned CodeLlama-13b model employing such methods. Our results indicate that the non-fine-tuned model performs poorly in 0-shot evaluations and underperforms compared to the fine-tuned model in 3-shot evaluations. We have updated Table 3 and Figure 4 to include these baseline results, providing a practical comparison that demonstrates the utility of our API Pack.

---

> ### Author Response · Authors · 2024-11-21
>
> * **[Minor] Base Model Selection: The selection of base models seems somewhat detached from practical application. API Pack functions as a specialized fine-tuning stage post-general SFT, targeting API call generation specifically. Therefore, selecting foundational SFT code models, like CodeLlama-Instruct or Magicoder, as discussed in Appendix A.8, would enhance the practical relevance and clarity of the paper’s contributions. Including more experiments with these models would substantiate the paper's claims.**
>     * Thank you for your feedback on our base model selection. We believe that whether to fine-tune a base model or an SFT model is a topic open to debate. In our paper, we chose to fine-tune on top of the base model to isolate the effect of the API Pack from the post-training processes used by different model providers. This approach allows us to assess the direct impact of the API Pack on code generation capabilities.
>     * We acknowledge that some practitioners might prefer to further fine-tune an SFT model. We leave this choice to those who wish to use the API Pack, allowing them to decide the best fine-tuning strategy for their specific applications.
>
> * **[Minor] Evaluation on General Code Generation Benchmarks: The paper does not discuss potential performance trade-offs on general code generation benchmarks, such as HumanEval, that could drop due to fine-tuning for API-specific tasks.**
>     * Thank you for raising the concern about potential performance trade-offs on general code generation benchmarks due to fine-tuning for API-specific tasks. We have conducted experiments to evaluate this, as now detailed in Section 5.5 in the revised manuscript, "Improving Code Models with API Pack."
>     * Our results, presented in Table 6, show that incorporating the API Pack with the Magicoder dataset to fine-tune the CodeLlama-13b model leads to a significant improvement of over 35.3% in API call code generation accuracy for Level 3 tasks in a 3-shot setting. Importantly, this improvement does not come at the expense of general coding performance. The model continues to perform well on benchmarks such as HumanEval+ and MBPP.
>     * These findings demonstrate that fine-tuning with the API Pack enhances API-specific capabilities while maintaining general coding performance, addressing your concern effectively.
>
>
>
> **Questions**
> * **How do the authors view the trade-offs between a RAG approach (with minimal or no fine-tuning) and the approach proposed here of fine-tuning on a large, static API dataset? Since APIs are continually updated, it seems that a RAG or search engine approach could offer more reliable, dynamic knowledge.**
> * **How does the static nature of API Pack compare in reliability and practical utility?**
>     * Thank you for your questions. We recognize that Retrieval-Augmented Generation (RAG) methods can provide dynamic, up-to-date API information without additional fine-tuning, which is beneficial as APIs evolve. However, our approach of fine-tuning on a large, static API dataset like API Pack embeds API knowledge directly into the model. This leads to more fluent and context-aware code generation without relying on external retrieval systems, which can introduce latency and dependencies. While RAG offers freshness, our method provides a self-contained model with consistent performance.
>     * The static nature of API Pack offers a stable and reliable knowledge base for code generation. Although it may not include the very latest API updates, it encompasses a wide range of commonly used APIs that remain relevant. This makes it practically useful for many applications. For scenarios requiring the most current API information, users can periodically update the API Pack or integrate our method with RAG approaches.

---

> ### Author Response · Authors · 2024-12-01
>
> Dear Reviewer NcGM
>
> Thank you sincerely for your thoughtful and thorough review of our manuscript, which has greatly improved its quality. As the discussion stage nears its conclusion, we kindly ask if you have any remaining questions or concerns. Any additional feedback would be greatly appreciated and will help us further enhance our work.
>
> Additionally, we would be grateful if you could consider adjusting the score in light of our revisions and clarifications, which were directly informed by your insightful feedback.
>
> Thank you again for your time and effort in reviewing our paper!

---

### Official Review · Reviewer_FEcU · 2024-11-07

**Soundness:** 4
**Presentation:** 3
**Contribution:** 4
**Rating:** 6
**Confidence:** 4

**Summary:**

The paper introduces API Pack, a large-scale dataset with more than a million instruction-API call samples. The authors discuss the aspects of API Pack making it unique and how it was constructed from public sources. Experiments are conducted to demonstrate: (1) Code LLM performance on API Pack under various RAG and evaluation settings (2) strong multi-lingual generalization on API Pack tasks from training a lot on one language and a little bit on others, (3) data-scaling properties hold when training on API Pack data, (4) comparisons with another training dataset, ToolBench.

**Strengths:**

- **Useful contribution.** The paper presents API Pack, which is a large-scale and multi-lingual instruction-API calling dataset. This can be used for both training and evaluations in multiple languages. In Table 1, the authors demonstrate how this is better along multiple dimensions than previous works.
- **Several detailed insights.** The paper is packed with insightful experiments. I particularly liked the experiments covering multi-lingual generalization — the designs of control experiments for this are interesting, and show that it is sufficient to train on a lot of data from one language and little data from other languages. There are other experiments, including ablations on RAG strategies and data volume used during training, which will also be valuable to the community.
- **Reproducibility.** Section 3, discussing the creation of API Pack, is extremely detailed and will be of great value to researchers looking to process data for similar tasks.

**Weaknesses:**

- **Lack of certain details.**
    - The baseline performance of non fine-tuned models on API Pack is missing (unless I missed it somewhere). Including this will greatly speak to the advantages of fine-tuning on API Pack.
    - For Section 5.3, how is the multi-lingual generalization being measured? Is it the same API invocation across different languages?
    - Could API Pack suffer from data leakage when evaluating public models trained on public sources?
- **Additional experiments.** While there are several experiments conducted, I am curious about a few more experiments. This is not a major concern however. While Section 5.5 does some comparison of API Pack with ToolBench, I believe that a more thorough evaluation requires training on API Pack and evaluation on ToolBench, Gorilla, etc. and also training on ToolBench, Gorilla, etc. and evaluating on API Pack. This would concretely identify where API Pack stands out and lacks relative to other API calling benchmarks out there.
- **Paper writing.** Overall the paper seems to be a bit verbose, and could benefit from making the writing more concise and crisp. For e.g., Section 5.3 is too wordy and a jumble of multiple findings. It would benefit the paper to organize such findings in a clearer way.
- **Evaluation metrics.** While string matching based metrics are being used for evaluations here and have been used in several previous works as well, I think it is time that we move on from these inaccurate metrics. We should resort to more robust metrics relying on static or execution analysis such as pass rate [1] or hallucination rate [2] for measuring correctness of code completions.
- **Nitpicks.**
    - Line 381 - “simple retrieval might be sufficient in this context” — I think it can also be the case that using a better re-ranker might help improve performance. For e.g., what’s the performance with the oracle retriever/re-ranker?
    - Figure 2
        - The legend(s) should be moved to make the results more readable
        - Is the “Expert” model in each subplot different? This should be clarified in the text/caption given all models have the same color
    - Figure 4 - it would be better for comparison if all the subplots had the same y-axis limits

[1] Chen, Mark, et al. "Evaluating large language models trained on code." *arXiv preprint arXiv:2107.03374* (2021).

[2] Jain, Nihal, et al. "On Mitigating Code LLM Hallucinations with API Documentation." *arXiv preprint arXiv:2407.09726* (2024).

**Questions:**

See notes above. I have translated some of the above in question format below:

1. What is the baseline performance of non-fine-tuned models on API Pack?
2. How is multi-lingual generalization being measured in Section 5.3? Is it the same API invocation across different languages?
3. Have you considered training on API Pack and evaluating on other benchmarks like ToolBench and Gorilla, and vice versa, to more thoroughly compare API Pack with existing datasets?
4. What is the performance with an oracle retriever/re-ranker in the context of the simple retrieval discussion?
5. Can you characterize if there are any data leakage concerns and/or what researchers should be mindful of when evaluating with API Pack?

---

> ### Author Response · Authors · 2024-11-21
>
> Thank you for your thoughtful and detailed feedback. We have carefully addressed your questions and updated the paper accordingly, with all changes highlighted in blue for clarity.
>
> **Questions**
> * **What is the baseline performance of non-fine-tuned models on API Pack?**
>     * Thanks for the question. We have added the non-finetuned baseline for CodeLlama-13b in the Table 3 and Figure 4 respectively. We found that the non-finetuned model performs poorly on 0-shot evaluations, and underperform finetuned model on 3-shot evaluations.
> * **How is multi-lingual generalization being measured in Section 5.3? Is it the same API invocation across different languages?**
>     * You are correct that we evaluate the same API invocation with different languages. We have revised Section 4 and Section 5.3 to enhance the transparency of our multi-lingual evaluation setting.
> * **Have you considered training on API Pack and evaluating on other benchmarks like ToolBench and Gorilla, and vice versa, to more thoroughly compare API Pack with existing datasets?**
>     * In Section 5.5, we described an ablation study where we trained on the ToolBench dataset on APIs and found that using the full set of API Pack led to better performance given the large data volume and diversity. We did not compared with Gorilla because it invokes APIs through model import instead of HTTP APIs as used in our paper.
> * **What is the performance with an oracle retriever/re-ranker in the context of the simple retrieval discussion? and in Line 381 - “simple retrieval might be sufficient in this context” — I think it can also be the case that using a better re-ranker might help improve performance. For e.g., what’s the performance with the oracle retriever/re-ranker?**
>     * Thank you for the suggestion. We have now added 3-shot oracle retriever evaluation results in Table 4. We found that oracle retriever indeed improves the intent and API call accuracies for both Mistral and CodeLlama. We thus have revised the discussion in Section 5.2 accordingly.
> * **Can you characterize if there are any data leakage concerns and/or what researchers should be mindful of when evaluating with API Pack?**
>     * Thank you for your feedback. For details on ethical considerations and potential concerns, please refer to Appendix A.1. Briefly, key points include ensuring datasets are updated to account for API evolution, verifying model outputs to mitigate hallucination risks, being mindful of data ownership and licensing, and recognizing that sensitive information is not included as placeholders are used in place of real argument values.
>
> **Comments about presentation**
> * **Overall the paper seems to be a bit verbose, and could benefit from making the writing more concise and crisp. For e.g., Section 5.3 is too wordy and a jumble of multiple findings. It would benefit the paper to organize such findings in a clearer way.**
>     * Thank you for your comments on the presentation. We have significantly revised the paper to improve clarity and conciseness, with the updates highlighted in blue for easier review. In particular, Section 5.3 has been reorganized to present the findings more clearly. We hope these changes address your concerns.
>
>
> * **Evaluation metrics. While string matching based metrics are being used for evaluations here and have been used in several previous works as well, I think it is time that we move on from these inaccurate metrics. We should resort to more robust metrics relying on static or execution analysis such as pass rate [1] or hallucination rate [2] for measuring correctness of code completions.**
>     [1] Chen, Mark, et al. "Evaluating large language models trained on code." arXiv preprint arXiv:2107.03374 (2021).
>     [2] Jain, Nihal, et al. "On Mitigating Code LLM Hallucinations with API Documentation." arXiv preprint arXiv:2407.09726 (2024).
>     * We appreciate the reviewer's suggestion on evaluation metrics. While execution-based metrics like pass rate and hallucination rate are indeed more robust, our dataset's focus on privacy protection (removing PII from API endpoints) makes execution-based evaluation difficult. We have acknowledged this limitation and cited papers [1, 2] in both our evaluation and limitations sections. For future work, we plan to explore enabling execution-based evaluation while maintaining privacy through synthetic API endpoints or sanitized test environments.

---

> > ### Comment · Reviewer_FEcU · 2024-12-02
> > **Reviewer Response**
> >
> > Thank you for your detailed comments!
> >
> > > Thank you for your feedback. For details on ethical considerations and potential concerns, please refer to Appendix A.1. Briefly, key points include ensuring datasets are updated to account for API evolution, verifying model outputs to mitigate hallucination risks, being mindful of data ownership and licensing, and recognizing that sensitive information is not included as placeholders are used in place of real argument values.
> >
> > This is very helpful information. However, my question was specifically on addressing concerns of data leakage. It would be great if that can be addressed to help readers know if their models may already have been trained on data from your benchmark.

---

> ### Author Response · Authors · 2024-11-21
>
> Following the previous comment:
>
> * Nitpicks
>     * **Figure 2 - The legend(s) should be moved to make the results more readable
>     Is the “Expert” model in each subplot different? This should be clarified in the text/caption given all models have the same color**
>     * **Figure 4 - it would be better for comparison if all the subplots had the same y-axis limits**
>         * Thank you for the detailed feedback. We have adjusted the legends in Figures 2 and 3 to enhance readability and clarified in the captions that the Expert models are specific to each programming language.
>         * Regarding the y-axis in Figure 4, we found that setting the same range made the trends in the plots less obvious. After careful consideration, we decided not to change the y-axis limits to best preserve the visual insights from the figure.

---

> ### Author Response · Authors · 2024-12-02
>
> ## Data Leakage
> Our dataset is designed for training purposes not to serve as a benchmark. However, we understand developers may need to easily verify what APIs are included in API Pack to avoid duplication with other data sources they use. We will include a list of all APIs in our appendix for easy review, and an example in our codebase to filter out instances based on API name (this is possible as all instances are annotated with the API name).
>
> ## Ethical considerations
> We have been working more on ethical considerations, in a recent analysis we utilized the Presidio library to scan our dataset for Personally Identifiable Information (PII) content. The Presidio library supports a wide range of entities, and we used all of them. We found that approximately 18.0% of the curl subset and 18.3% of the entire dataset, which encompasses data from all languages, were annotated as having PII issues. To verify these findings, we manually inspected 1,000 instances from the curl subset that had been flagged with PII content. Our manual review revealed the following:
>
> - Public URLs (e.g., google.com, nestle.com, unilever.com, https://www.nytimes.com/2020/03/21/arts/d-nice-instagram.html)
> - API Name (e.g., the MaintenanceManagementClient API, the Gathering API)
> - Locations (e.g., Seattle, UK, New York City, US)
>
> In our view, the examples manually checked are misclassifications. First, our instructions should explicitly mention the API name by design. Second, location data, in its raw form, does not constitute Personally Identifiable Information (PII) unless it discloses a real person's birthplace or home address. Our manual review of the examples found no matches that fit this description. As for URLs, all the ones mentioned in the 1,000 samples are publicly accessible. Moreover, many of them reference the APIs themselves, which were deliberately exposed by the API owners in their documentation. To ensure transparency and cater to different preferences, we will add a column to our dataset to label all instances identified as having potential PII concerns in the instructions, regardless of whether they are false positives. This way developers can easily filter them out if that is their preference. We will also update the paper to describe the PII analysis performed.
>
> In relation to PII concerns in the response, it is important to note that the data utilized in the response originated from public-technical documentation websites, rather than real-world data crawled from the web. These websites employ placeholders to symbolize the arguments that developers should incorporate into their code.
>
> Below, we show an example of an instance flagged as containing PII. Note that the url in the instruct is public, and that the url in the response is the API url. Please also note that a placeholder is used to represent the arguments that developers should pass for the call to work, real information is not used.
>
> ### Example
>
> #### ---- Instruction ----
> Could you help me extract essential and comprehensive information from a news article using the News Media-Extracting Essential Information From News Articles Url-API? For example, this API can be used to extract the following details from the URL <https://www.latimes.com/california/story/2020-05-20/defying-state-order-thousands-of-pastors-say-they-plan-to-hold-in-person-services-for-the-pentecost>: date, author, description, summary, natural language processing information, entities (such as PEOPLE, PLACES, THINGS, EVENT, ORGANIZATION), and related images. By making a request to the /extractfromArticle endpoint of the News Media-Extracting Essential Information From News Articles Url-API with the given URL, I can obtain all this essential and comprehensive information in a structured format.
>
>
>
>
>
>
>
>
>
>
>
>
> #### ---- API Call ----
> ```
> curl --request GET \
>   --url 'https//web.pregnya.comhttps://extracting-essential-information-from-news-articles-url.p.rapidapi.com/extractfromArticle?url=SOME_STRING_VALUE' \
>   --header 'X-RapidAPI-Host: SOME_STRING_VALUE' \
>   --header 'X-RapidAPI-Key: SOME_STRING_VALUE'
> ```

---

> ### Author Response · Authors · 2024-12-02
>
> Thank you for highlighting this concern. Addressing data leakage is crucial. Please refer to the official comments above for our analysis and the steps we are taking to address these issues. Let us know if you have further questions or need additional clarification.

---

### Author Response · Authors · 2024-12-03

We sincerely thank all reviewers for their thoughtful feedback that helped us improve our work. The reviewers highlighted key strengths of our paper: the value of API Pack as a large dataset for API learning (Reviewers FEcU and FeUP), our thorough experiments (Reviewers bcCN and NcGM), and our careful data collection process (Reviewers FEcU and FeUP).

Based on their suggestions, we have made important improvements: we added new baseline experiments, clarified our methods and metrics, expanded related work, and carefully addressed data quality concerns. We also added new analysis in the appendix to show how well our approach works in terms of instruction data quality.

The reviewers' feedback has helped us explain our work more clearly and make it more useful to others. We look forward to sharing API Pack with the research community.

Thank you again to all reviewers for their time and expertise in helping us strengthen this paper.

---

### Meta-Review · Area_Chair_D6Ja · 2024-12-13

**Metareview:**

The paper presents API Pack, a large-scale dataset comprising over one million instruction-API call samples. The authors highlight the unique aspects of API Pack and detail its construction from publicly available sources. Experiments demonstrate several key findings: (1) the performance of code LLMs on API Pack under various retrieval-augmented generation (RAG) and evaluation settings, (2) strong multilingual generalization on API Pack tasks achieved by training extensively in one language while using limited data from others, (3) the retention of data-scaling properties when training on API Pack data, and (4) comparisons with another training dataset, ToolBench.

The reviewers generally agree that this is a strong paper; however, they raised some concerns regarding the comparisons with related work and the validity of the benchmark. The authors should carefully revise the paper in accordance with the reviewers' comments.

**Additional Comments On Reviewer Discussion:**

The reviewers generally agree that this is a strong paper; however, they raised some concerns regarding the comparisons with related work and the validity of the benchmark. The authors should carefully revise the paper in accordance with the reviewers' comments.

---

### Decision · Program_Chairs · 2025-01-22

Accept (Poster)